

# Formation of secondary organic aerosols from the ozonolysis of dihydrofurans.

Diaz de Mera, Yolanda[1]; Aranda, Alfonso[1]; Bracco, Larisa[2]; Rodriguez, Diana[3]; Rodriguez, Ana[3].

[1]Universidad de Castilla-La Mancha. Facultad de Ciencias y Tecnologias Quimicas. Avenida Camilo José Cela s/n, 13071. Ciudad real. Spain.

[2]Instituto de Investigaciones Fisicoquímicas Teóricas y Aplicadas (INIFTA), Departamento de Química, Facultad de Ciencias Exactas, Universidad Nacional de La Plata, CONICET, Casilla de Correo 16, Sucursal 4, La Plata 1900, Argentina.

[3]Universidad de Castilla-La Mancha. Facultad de Ciencias Ambientales y Bioquimica. Avenida Carlos III s/n, 45071 Toledo, Spain.

*Correspondence to:* Alfonso Aranda (alfonso.aranda@uclm.es)

**Abstract.** In this work we report the study of the ozonolysis of 2,5-dihydrofuran and 2,3-dihydrofuran and the reaction conditions leading to the formation of secondary organic aerosols. The reactions have been carried out in a Teflon chamber filled with synthetic air mixtures at atmospheric pressure and room temperature. The ozonolysis only produced particles in the presence of $SO_2$. Water vapour has no effect on the production of secondary organic aerosol in the case of 2,5-dihydrofuran while it reduces the particle number and particle mass concentrations from the 2,3-dihydrofuran ozonolysis. The water and $SO_2$ rate constants ratio for the 2,3-dihydrofuran Criegee intermediate was derived from the SOA yields in experiments with different relative humidity values, $k_{H2O}/k_{SO2}= (9.8\pm3.7)\times10^{-5}$.

The experimental results show that $SO_3$ is not an intermediate in the formation or growth of new particles in contrast to the data reported for other Criegee intermediates/$SO_2$ reactions. For the studied reactions, $SO_2$ behaves as a catalyst in the production of condensable products.

Computational calculations show that the stabilised Criegee intermediates from the ozonolysis reaction of both 2,5-dihydrofuran and 2,3-dihydrofuran may react with $SO_2$ resulting in the regeneration of $SO_2$ and the formation of organic acids.

## Introduction.

Recent years have seen a growing interest in the atmospheric mechanisms leading to the formation of secondary organic aerosols, SOA (Chan et al., 2010). In this sense, ozonolysis of terpenes has been reported as a potential source of new particles under natural conditions (Saathoff et al., 2009; Sipilä et al., 2014; Newland et al., 2015a). Likewise, the fast reactions of other alkenes with ozone can contribute to the production of non-volatile species that could condense and contribute to the total mass of pre-existing particles or even induce nucleation events in both rural and urban atmospheres.





As it is known (Johnson and Marston, 2008), ozone adds to the double bond giving an energy-rich primary ozonide which promptly decomposes giving a carbonyl molecule and an excited carbonyl oxide reactive intermediate (Criegee Intermediate, CI). For cyclic alkenes, the ozonolysis just opens the cycle giving larger CI compared to the equivalent linear alkenes. Furthermore, the carbonyl functional group is included in the CI molecule. Thus, products with higher molecular weight, lower

volatility and higher potential capacity to contribute to SOA formation are expected for chemicals with endocyclic double bonds. The excited CI intermediate can be stabilised by collisions with gas molecules or undergo fragmentation or unimolecular rearrangement (Anglada et al., 2011). Then, the stabilised CI (sCI) can react with other molecules in the gas phase such as $H_2O$, $HO_x$, aldehydes, organic acids, $RO_2$, $NO_x$, etc (Vereecken et al., 2012). The reaction with water molecules is expected to be one of the main fates of sCI, giving $\alpha$-hydroxi-hydroperoxides (Ryzhkov and Ariya 2004; Anglada et al.,

10   2011).

On the other hand, experimental results show that $SO_2$-sCI reactions are a potential source of SOA. Very recently, it has been found that a significant fraction of ground level sulfuric acid originates from the oxidation of sulphur dioxide by sCI to $SO_3$ (Mauldin et al., 2012) and, thus, several studies have been carried out on the reactions of Criegee intermediates with $SO_2$ (Boy et al., 2013; Berndt et al., 2014a; Stone et al., 2014; Newland et al., 2015b; Liu et al., 2016). The relative contribution of $SO_2$

and water vapour to the CI removal in the atmosphere depends on the CI structure (Berndt et al., 2014a; Stone et al., 2014) and may have an important effect on the SOA formation yields.

In this work we report the study of the ozonolysis of 2,5-dihydrofuran (2,5-DHF) and 2,3-dihydrofuran (2,3-DHF) under variable concentrations of water vapour and $SO_2$.

2,5-DHF + $O_3$ →… SOA   (R1)

2,3-DHF + $O_3$ →… SOA   (R2)

Very recently, it has been found that different dihydrofurans may be involved in the SOA formation from alkane photooxidation (Loza et al.,2014; Zhang et al., 2014). Alkoxy radicals are initially generated from the reaction with OH. They can subsequently isomerize into $\delta$-hydrocarbonyl intermediates which undergo cyclization and dehydration producing dihydrofurans (Martín et al., 2002).

2,3-dihydrofuran is also found in the atmosphere due to the emissions from biomass burning (Lemieux et al., 2004) and is included in atmospheric chemistry models aimed at the emissions of aerosols (Freitas et al., 2011). Furthermore, during the last few years, different oxygenate chemicals are being tested as fuel additives or components of biofuels. Thus, furans and derivatives are potential second-generation biofuels since they could be produced from fructose and glucose. (Román-Leshkov et al., 2007). In this sense, the ignition characteristics of 2,5-DHF have been reported and compared to those of other

heterocyclic compounds (Fan et al., 2016).

A previous work has reported the tropospheric oxidation study of 2,5-DHF and 2,3-DHF (Alwe et al., 2014). The measured rate constants with ozone are: 1.65 and 443.2 × $10^{-17}$ cm$^3$ molecule$^{-1}$ s$^{-1}$ for 2,5-DHF and 2,3-DHF, respectively. Alwe et al.



(Alwe t al., 2014) show that the dominant pathway of tropospheric degradation of 2,5-DHF is the reaction with OH, whereas for 2,3-DHF is the reaction with $O_3$. So, ozonolysis of 2,3-DHF is a potential source of CI in the troposphere. Although the reaction of 2,5 DHF with ozone is slower, the result for this unsaturated cyclic ether may serve as reference for other substituted dihydrofurans.

Up to now, no previous studies have been carried out to assess the potential capacity of the title compounds to generate condensable matter and SOA. Thus, in this work we study the ozonolysis reactions of 2,5-DHF and 2,3-DHF following the conditions that lead to the formation of new particles. The effect of water vapour and $SO_2$ concentrations during the process are also studied and discussed. The experimental work is supported by theoretical calculations to explore the key steps involved within the reaction mechanism leading to SOA.

**2. Methods and instrumentation:**

**2.1. Experimental set-up and methods.**

A schematic diagram of the experimental system is shown in Fig. 1. A 200L capacity FEP collapsible chamber has been used to carry out most of the experiments under atmospheric pressure at 297±1K. The reactants were sequentially sampled in a volume-calibrated glass bulb (1,059L) using capacitance pressure gauges, MKS 626AX, 100 and 1000Torr full scale, and

then flushed to the reactor using synthetic air through a mass flow controller, MFC 1179BX, MKS). The concentrations of each substance were calculated from their partial pressure in the bulb and the dilution factor considering the glass bulb volume and the final volume of the Teflon reactor.

Ozone was produced by an ozone generator, BMT Messtechnik 802N, fed with pure oxygen. The ozone/oxygen mixture was introduced simultaneously to the sampling bag and to a 19.8 cm long quartz cell to measure the absorption of radiation at

255nm using a UV-vis Hamamatsu spectrometer, C10082CAH, with a 1 nm resolution. The ozone concentration in the bulb was then calculated from the known value of the ozone absorption cross section. Further dilution of ozone enabled the required range of ozone concentrations in the reactor.

Water vapour was generated using a glass bubbler placed just before the reactor inlet. The amount of water required for a given relative humidity was calculated and injected in the bubbler. All the sampled water was then evaporated and flushed inside the

reactor through the flow of synthetic air.

The certified water impurity content in the synthetic air used was below 2ppm. Although it is generally negligible, it could be significant if water molecules were involved in fast reactions. To properly asses the effect of water in the studied reactions, some experiments were carried out under completely dry conditions. In such experiments, the flow from the air cylinder was passed through a trap containing molecular sieve 5A (Supelco) cooled down to 157K with an ethanol-liquid nitrogen slush

bath. Some experiments were also carried using a liquid nitrogen trap and checking that no condensed oxygen was visible in the trap.

The sequence of reactants entering the Teflon bag was as follows. First, 2,5-DHF (or 2,3-DHF) was introduced, then the OH

scavenger (cyclohexane), then $SO_2$ and finally ozone. In the experiments carried out with water vapour, water was introduced

in the bubbler at the beginning, being evaporated by the air carrying the DHF, the scavenger and $SO_2$.

The injection of ozone constitutes the ignition of the 2,5-DHF (or 2,3-DHF)–ozone reaction.  To completely wash away the

ozone from the bulb and transport it to the reactor, a flow of synthetic air was required for 45 seconds. During this time period

the Teflon reactor was shaken to accelerate the mixing of reactants.

The formation of particles was followed continuously by a CPC particle counting 3775 TSI and, at given times, samples were

derived to a fast mobility particle sizer (FMPS 3091,TSI) to obtain the particle size distribution and total mass of particles with

diameters within the range 5.6–560 nm (Aranda et al., 2015). The total aerosol mass concentration was calculated from the

measured particle size distribution assuming unit density and spherical particles.

Prior to each experiment the Teflon bag was repetitively filled and emptied with clean synthetic air to remove particles

remaining from previous experiments. The process was continued until the level of particles was below 1 #cm$^{-3}$.

For some experiments, a fluorescence $SO_2$ analyser, Teledyne Instruments 101-E, was coupled to the reactor to measure the

$SO_2$ concentration profiles during the experiments. Previous runs with only $SO_2$ /air samples in the reactor showed that $SO_2$

wall losses were negligible. Additionally, no interference in the $SO_2$ fluorescence signal from the rest of co-reactants was

observed in the range of concentrations used in this study.

**2.2. Computational methods.**

  All the calculations were performed with the Gaussian09 program package (Frisch et al., 2009). The BMK formulations

(Boese and Martin, 2004) of the density functional theory (DFT) combined with the Pople triple split-valence basis set 6-

311++G(3df,3pd) were employed. In all cases, the structural parameters were fully optimized via analytic gradient methods.

The Synchronous Transit-Guided Quasi-Newton (STQN) method was employed for locating transition structures. For a

better estimation of energies, single point calculations at ab initio level CCSD(T) were performed using the  6-311G(d,p)

basis set (Cizek, 1969; Bartlett 1989).

**2.3 Reagents**.

All the substances used were of the highest commercially available purity. Liquid reagents were purified by successive trap

to trap distillation. 2,5-DHF 97%, Aldrich; 2,3-DHF 99%, Aldrich; $SO_2$ 99,9%, Fluka; cyclohexane 99.5%, Sigma-Aldrich;

synthetic air 3X, Praxair; $O_2$ 4X, Praxair.

**3. Results and discussion.**



### 3.1. 2-5-Dihydrofuran.

### 3.1.1. Conditions for SOA formation.

Cyclohexane in excess was used as OH radical scavenger in concentration ratios so that >95% of OH formed in the reactions was removed (Ma et al., 2009). Different series of experiments were carried out to characterize the formation and growth of

particles originated from the reaction of ozone with 2,5-DHF.

In Fig. 2 we show a typical experiment carried out with 0.5ppm of 2,5-DHF, 1ppm of $O_3$ and 0.5ppm of $SO_2$ with zero relative humidity (RH) initial concentrations. Since new particles are readily formed due to this reaction, low volatile substances must be produced and over-exceed several times their saturation vapour pressure, initiating homogeneous nucleation (Kelving effect) (Pruppacher et al., 1996). The experimental data obtained for the particle number concentration are presented together

with the simulated profile of 2,5-DHF (from the known initial concentrations and the gas-phase rate constant (Alwe et al., 2014) and the total concentration of consumed 2,5-DHF, Fig. 2. Small particles with diameters above 4nm were quickly detected by the CPC. A maximum in the particle number concentration (PNC) was observed around 8 minutes and then PNC progressively decreased with time mainly due to coagulation of particles and wall losses. For reaction times longer than 2 hours and PNC below $1x10^4$#cm$^{-3}$ its decrease may be attributed solely to wall losses. The plot of Ln(PNC) against time for

different experiments provided linear fittings and an average $K_w$ of $(5\pm2)\times10^{-5}$s$^{-1}$.

The total mass concentration (or particle mass concentration, PMC) of measured aerosol is also shown in Fig. 2 multiplied by a factor scale since the units are different to the rest of magnitudes. The PMC profile arises after the nucleation event and continues growing until approximately 30 minutes. Then a low decrease of the mass concentration is observed with time. The profile of total concentration of consumed 2,5-DHF is very similar to that of the mass concentration. This fact suggests that

reaction (R1) is the limiting step in the production of condensable matter. Beyond 30 minutes the mass concentration decreases, mainly due to wall losses, although 2,5-DHF is still reacting. Better fits of these two profiles are obtained when the wall rate constant is considered.

A series of experiments with lower concentrations of reactants was conducted to characterize the nucleation conditions. Thus, for example, in Fig. 2 we can also see the profile of the particle number concentration for initial concentrations of 2,5-DHF,

ozone and $SO_2$ 0.1, 0.2 and 0.1 ppm respectively. In such experiment no particles were detected before 3 minutes total time. For such reaction time and conditions, the reacted concentration of 2,5-DHF amounts $3.5x10^{10}$molecule cm$^{-3}$. From the average of the different experiments, we estimate that an upper limit concentration of $3.5\pm1x10^{10}$molecule cm$^{-3}$ is required for the direct gas-phase product from reaction (R1) to originate the nucleation event.

### 3.1.2. Effect of SO₂ and water.





When the experiments were carried out in the absence of $SO_2$, no particles were observed. On the other hand, when $SO_2$ was added, an increase in PNC and in the total condensed mass is observed for increasing concentrations of $SO_2$, Fig. S1 (supplementary material). Even for relatively low $SO_2$ concentrations (0.2ppm) high concentrations of particles (above $10^5$ #$cm^{-3}$) were measured. This series of experiments was conducted under completely dry conditions. So for these experiments,

the reaction of $SO_3$ with water cannot be responsible for the formation of particles, showing that there is a "dry channel" able to produce organic non-volatile species able to condense.

Some experiments were conducted also following the temporal profile of $SO_2$. The $SO_2$ concentration remained constant during these experiments showing that, although sulphur dioxide participates in the mechanism leading to particles, it is released again as free gas-phase $SO_2$.

Under atmospheric conditions, reactions with water vapour is expected to be one of the main fates of CI intermediates (Ryzhkov and Ariya, 2004). For 2,5-DHF, Fig. S2 shows that increasing the RH within the range 0 to 40% had no significant effect in the measured values of both PNC and PMC. Thus, concerning the potential competing reactions of the sCI with water vapour and $SO_2$, the results suggest that water reaction contribution is negligible. The fact that water has no effect on the production of SOA also suggests that $SO_3$ (which would react with water) is not present.

Concerning the effect of ozone and 2,5-DHF initial concentrations, different series of experiments were carried out. As discussed above, the ozone reaction with 2,5-DHF was the rate limiting step in the production of SOA. Thus, the increase of $[O_3]$ or [2,5-DHF] led to the acceleration of the process and the increase of the measured PNC and PMC. See supplementary information for further details.

### 3.2. 2,3-Dihydrofuran.

**3.2.1. Effect of $SO_2$.**

In Fig. 3 and S6 we show the results for a typical experiments carried out with 1 ppm of ozone and 0.5 ppm of 2,3-DHF under dry conditions. When the experiments were carried out in the absence of $SO_2$, no particles were observed. On the other hand, when $SO_2$ was added as co-reactant (see for example the profile of the experiment with 0.5 ppm initial concentrations of $SO_2$ and 0% HR) particles suddenly originated after approximately 1 minute after the introduction of ozone in the reactor. The

number of particles still grew until approximately 5 minutes and then decreased during the rest of the experiment.

In Fig. 3 we can also see the simulated profile for [2,3-DHF]. The ozone-2,3-DHF reaction is very fast and 2,3-DHF is consumed within the first minute of the experiment while the mass of condensed matter starts growing after nucleation and continues beyond 20 minutes. These results show that the initial ozone-2,3-DHF reaction itself is not the rate limiting step in the mechanism leading to new particles, in contrast to the results for 2,5-DHF.

For experiments with increasing concentrations of $SO_2$, nucleation was found at shorter times and higher PNC values were obtained. The total mass and diameter of particles also increased with $SO_2$, Fig. 3 and Fig. S7. For the smaller $SO_2$





concentration, (0.05ppm) a clear delay of the burst of nucleation is observed, Fig. 3. These results suggest that $SO_2$ is involved within the first steps of the mechanism driving to the nucleation event. For such reason the potential secondary ozonide formed from the sCI reaction with $SO_2$ has been characterized, see below.

For reaction (R2), as in the case of reaction (R1), some experiments were conducted following the temporal profile of $SO_2$.

The $SO_2$ concentration remained constant during these experiments. The facts that new particles are generated only in the presence of $SO_2$ and that $SO_2$ does not decrease during the experiments suggest that sulphur dioxide behaves as a catalyst in the production of condensable products leading to particles.

### 3.2.2. Effect of 2,3-DHF and ozone initial concentrations.

Two series of experiments were carried out changing the ratio of 2,3-DHF over ozone for a fixed concentration of $SO_2$ and in

the absence of water vapour. The results are summarized in Table 1. When 2,3-DHF was in excess over ozone, it lead to a fast consumption of ozone (during the first minute). As show in the table, for a fixed low initial ozone concentration, the higher the concentration of 2,3-DHF, the lower the number concentration. Nucleation in these experiments (hardly observed) was also almost instantaneous but the PNC significantly decreased (at least two orders of magnitude) compared to the reference experiments (stoichiometric conditions) and the total mass fell to virtually zero. For those experiments, a significant difference

was observed in the number concentration data from the FMPS and the CPC. The FMPS hardly detected particles (they must have diameters above 5.6nm to be detectable) while the the CPC can detect smaller particles (down to 4 nm). The results showed that the particles did not grow when ozone was the stoichiometry limiting reactant. Thus, further ozone may be involved after the initial $O_3$-2,3-DHF reaction to produce SOA. These results are also consistent with the series of experiments with ozone in excess over 2,3-DHF. With ozone in excess, the particle number concentration and the total mass of particles

increased with the ozone concentration. As found in Table 1 or Fig. S8, the size of the particles also grew as the initial ozone concentration was increased.

### 3.2.3. Effect of water vapour.

Two series of experiments have been carried out in this work to study the effect of water. First, for the experiments conducted in the presence of water (RH from 0 to 50%) and in the absence of $SO_2$, no particle formation was observed. So, if water reacts

with the 2,3-DHF CI, this reaction does not produce particles in the absence of $SO_2$, Table S1 (supplementary material). In the second series of experiments $SO_2$ was also present, being all the experiments carried under the same initial concentrations of 2,3-DHF, ozone and $SO_2$ with HR ranging from 0 to 50%. As shown in Fig 4 and Fig. S9, the increase in RH reduced the particle number concentration and the mass of formed SOA. Figure 4 shows the results for two experiments, under dry conditions and under 50% RH. In the presence of 50% HR, the particle number concentration was significantly lower and the





measured mass concentration fell from 13 to approximately 3 µg cm$^{-3}$. Concerning the growth of particles, their final diameters were also smaller in the presence of water.

These effects may be due to the reaction of water with the Criegee intermediate which would compete with the sCI reaction towards $SO_2$. From previous studies, (Anglada et al., 2011) this reaction is expected to proceed through the addition of the

water oxygen atom to the carbon atom in the carbonyl oxide and the simultaneous transfer of one hydrogen atom from the water molecule to the terminal O atom in the CI. The expected products are α-hydroxi-hydroperoxides that can subsequently decompose or react with other atmospheric species.

If we take the dry experiments (0% HR) as reference, then we can assume that for the experiments carried out in the presence of water the decrease in mass concentration, $\Delta_{PMC}$, is due exclusively to the reaction of water molecules with the sCI. Water

would competitively consume sCI, with a kinetic rate constant $k_{H2O}$, against the reaction with $SO_2$, with a kinetic rate constant $k_{SO2}$. Thus, the ratio $(k_{H2O}.[H_2O])/(k_{SO2}.[SO_2])$ can be obtained from the ratio $\Delta_{PMC}$ /PMC, that is, from the drop in the mass concentration values measured at a given relative humidity ($\Delta_{PMC}$ drop is attributed solely to the reaction with $H_2O$) and the actual mass concentration obtained in such conditions (PMC, attributed to the reaction with $SO_2$). Thus, the ratio for the rate constants of the 2,3-DHF Criegee intermediate with water vapour and $SO_2$ was obtained plotting $\Delta_{PMC}$ /PMC versus [$H_2O$]

for constant and known $SO_2$ concentration (Fig. S10), $k_{H2O}/k_{SO2}= (9.8\pm3.7)\times10^{-5}$. This ratio is similar to other reported values for different CI, (Berndt et al., 2014a). This result may be used to assess different atmospheric conditions. For example, for relatively dry conditions (20% HR) in a polluted atmosphere with 75 ppbs of $SO_2$ (the current 1 hour NAAQS standard) at 25ºC, approximately 11% of the sCI produced in the ozonolysis of 2,3-DHF would react with $SO_2$ with potential to yield new particles and nucleation events. In the lower troposphere this source is not unique in the sense that other pollutants would be

simultaneously producing condensable products. Nucleation under real atmospheric conditions is due to the cumulative sources of non-volatile species and so these events are expected to generate under lower concentrations than those found in the laboratory for the individual sources.

### 3.3. Theoretical insights.

Recent studies have revealed that sCI may react with $SO_2$ and other trace gases several orders of magnitude faster than assumed

so far (Welz et al., 2012). Therefore sCI reactions have emerged as a potential source of tropospheric sulphate to be assessed in the predictions of tropospheric aerosol formation. Even though the atmospheric fate of the smaller carbonyl oxides is mainly dominated by the reaction with water molecules or water dimers, (Berndt et al., 2014a; Chao et al., 2015), the reactivity of sCIs is strongly dependent on the structure.

Previous theoretical studies have shown that the first step of the sCI with $SO_2$ is the formation of a cyclic secondary ozonide

(SOZ) that can undergo decomposition to the corresponding carbonyl and $SO_3$ or isomerization involving a 1,2-H-shift leading





to an organic acid and $SO_2$ (Jiang et al., 2010; Kurten et al., 2011). The release of $SO_3$ would produce $H_2SO_4$ in the presence of water molecules. Vereeken et al. (Vereeken et al., 2012), also described the formation of a stable ester sulfinic acid from the opening of the SOZ.

$$R_1R_2COO + SO_2 \rightarrow R_1R_2CO + SO_3 \qquad\qquad (R3)$$

$$\rightarrow R_1COOH + SO_2 \quad (\text{only if } R_2 = H) \qquad\qquad (R4)$$

$$\rightarrow R_1C(O)OS(O)OH \quad (\text{only if } R_2 = H) \qquad\qquad (R5)$$

In this sense we have studied the transition states for both 2,5-DHF and 2,3-DHF Criegee intermediates reactions with $SO_2$. Figure 5a shows a schematic representation of the general potential energy surface for all the studied reactions. Figure 5b displays the optimized geometries of the stationary points for all reactants, products and stationary states obtained at BMK/6-311++G(3df,3pd) level of theory.

Concerning the reaction mechanism, In the case of 2,5-DHF reaction with $O_3$, only one Criegee intermediate (sCI1) may be formed. On the other hand, due to the asymmetry of 2,3-DHF, two possible Criegee intermediates can be obtained (sCI2 and sCI3) from its reaction with $O_3$. For all these intermediates, there are two possible configurations, syn and anti. The latter, anti, were used in all cases to perform the potential energy surface calculations since they were more stable than the syn isomers in 1.08, 3.67 and 3.65 kcal mol⁻¹, respectively.

The reaction of sCI1 with $SO_2$ first evolves to the formation of an intermediate adduct M1, which then may react through two channels. On the one hand, through the transition state TS1.1, $SO_3$ and an aldehyde are formed. If the adduct M1 evolves via the transition state TS1.2, $SO_2$ is regenerated and an organic acid is produced. Due to the fact that both transition states lie below the energy of reactants, the most favourable reaction results in the regeneration of $SO_2$ and the organic acid with reaction energy of -117.58 kcal mol⁻¹. Table 2 shows the energies (relative to reactants sCI and $SO_2$) for the stationary species found. For the case of 2,3-DHF, the same mechanism is observed in the reactions of sCI2 sCI3 with $SO_2$. First the intermediate adducts, M2 and M3, are generated and then they evolve through two possible reaction pathways: through the transition state TS2.1 (or TS3.1) or transition state TS2.2 (or TS3.2) resulting in production of $SO_3$ and an aldehyde or $SO_2$ and an organic acid respectively. In both cases, similarly to what has been described previously for sCI1, the regeneration of the $SO_2$ and an organic acid is the more favorable pathway since the intermediate states lie below the energy of reactants. The reaction energies are -116.39 and -115.45 kcal mol⁻¹ for the organic acids 2 and 3, respectively, Table 2.

The theoretical results are consistent with the experimental findings described previously: $SO_3$ was not released and $SO_2$ concentrations remained unchanged during the laboratory experiments. So, reactions (R3) and (R5) may be ruled out as potential sources of new particles from the ozonolysis of 2,5-DHF and 2,3-DHF. On the other hand, both theoretical and experimental results show that SOA formation may generate from reaction (R4) for both 2,5-DHF and 2,3-DHF. Concerning



the organic acids from reaction (4),"organic acids 1", "2" and "3": $HC(O)CH_2OCH_2C(O)OH$ (formylmethoxyacetic acid), $HC(O)OCH_2CH_2COOH$ (3-formyloxypropanoic acid) and $HC(O)CH_2CH_2OCOOH$ (2-formyl-etoxyformic acid), Fig. 5, there are no vapor pressure data available in the literature. Similar chemicals like $HC(O)OCH(CH_3)COOH$ (2-formyloxy-propionic acid and $CH_3C(O)OCH_2COOH$ (acetoxyacetic acid) are solids at room temperature and have high melting and boiling points

(Zanesco, 1966; Buckingham and Donaghy, 1982). In this sense, low volatilities are also expected for the acids possibly generated from 2,5-DHF and 2,3-DHF, which could be the species responsible for the nucleation events.

**4. Conclusions.**

2,5-DHF and 2,3-DHF show a different behaviour during the ozonolysis showing that the reactivity of the CI is clearly dependent on the structure. For 2,5-DHF the mass increase of particulate matter correlates in time with the total

concentration of reacted alkene. On the other hand, for 2,3-DHF, the particle mass concentration rises well after the total consumption of this cyclic alkene. Furthermore, no effect of RH on the production of SOA was found for the ozonolysis of 2,5-DHF while the water vapour inhibits the SOA production from to the ozonolysis of 2,3-DHF.

As found in this work, the experimental results for the reaction of ozone with 2,5-DHF show that the presence of small amounts of $SO_2$ is required to generate SOA. Nevertheless, the main atmospheric fate of 2,5-DHF is the reaction with OH radicals and,

so, a small contribution to new particle formation in the atmosphere is expected from reaction (R1). Nevertheless, the information reported in this work contributes to the sparse current knowledge of the reactivity of CI intermediates with $SO_2$ and $H_2O$ and so it may be helpful in the understanding of the behaviour of other cyclic alkenes.

On the other hand, reaction (R2) is dominant over the OH reaction in the troposphere and, as it has been found in this work, the ozonolysis of 2,3-DHF leads to the formation of SOA in the presence of $SO_2$. The results show that under increasing

amounts of water vapour the yields of SOA decrease, suggesting the competing reactions of the sCI with $SO_2$ and water molecules.

The experimental findings reported in this work show that $SO_2$ participates as a catalyst in the in the production of new particles from the ozonolysis of both 2,5-DHF and 2,3-DHF. Thus, for these alkenes, the oxidation of $SO_2$ to $SO_3$ through reaction with the corresponding sCI must be ruled out, contrary to the results found for smaller carbonyl oxides (Berndt et al., 2014b). Since

$SO_3$ is not produced from reactions (1) and (2), $H_2SO_4$ can not be the species responsible for nucleation. Likewise, the pathway producing ester sulfinic acids (reaction (5)) can be ruled out for 2,5-DHF and 2,3-DHF. Thus, the organic acids produced through reaction (4) are expected to be the key species in the formation of SOA.

**Acknowledgments.**



This work was supported by the Spanish Ministerio de Ciencia e Innovación (project CGL2014-57087-R) and by the

University of Castilla La Mancha (projects GI20152950 and GI20163433).

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



**Table 1.** Initial concentrations of reactants and SOA data for the ozonolysis reaction of 2,3-DHF. PNC, PMC and diameter

reported values are the maximum data registered for each magnitude during the experiment.

| 2,3-DHF (ppm) | $O_3$ (ppm) | $SO_2$ (ppm) | PNC (#/$cm^3$) | PMC ($\mu g/m^3$) | Diameter (nm) |
|---|---|---|---|---|---|
| 0.2 | 0.5 | 0.5 | $5.7 \times 10^6$ | 2 | 25 |
| 0.5 | 0.5 | 0.5 | $5.8 \times 10^5$ | 1.9 | 30 |
| 1 | 0.5 | 0.5 | $3.0 \times 10^4$ | 0 | - |
| 2 | 0.5 | 0.5 | $1.0 \times 10^3$ | 0 | - |
| 3 | 0.2 | 0.5 | 27 | 0 | - |
| 3 | 0.5 | 0.5 | 47 | 0 | - |
| 3 | 1 | 0.5 | $5.3 \times 10^3$ | 0.12 | 80 |
| 0.5 | 1 | 0.5 | $1.70 \times 10^6$ | 10.1 | 40 |
| 0.5 | 2 | 0.5 | $2.10 \times 10^6$ | 18.9 | 50 |
| 0.5 | 3 | 0.5 | $3.30 \times 10^6$ | 33.5 | 50 |
| 0.5 | 4 | 0.5 | $3.40 \times 10^6$ | 38 | 60 |

5    **Table 2.** Relative energies to reactants sCI and $SO_2$ ($\Delta E + ZPE$) in kcal $mol^{-1}$ for stationary points of the all reaction.

Energies were obtained at CCSD(T)/6-311G(d,p) level of theory and zero point energies were obtained at BMK/6-

311++G(3df,3pd) level of theory.

| | A | B | C | B´ | C´ |
|---|---|---|---|---|---|
| sCI1 | -30.39 | -12.58 | -59.58 | -8.42 | -117.58 |
| sCI2 | -28.01 | -11.32 | -55.87 | -9.40 | -116.39 |
| sCI3 | -26.61 | -10.32 | 69.28 | -6.07 | -115.45 |





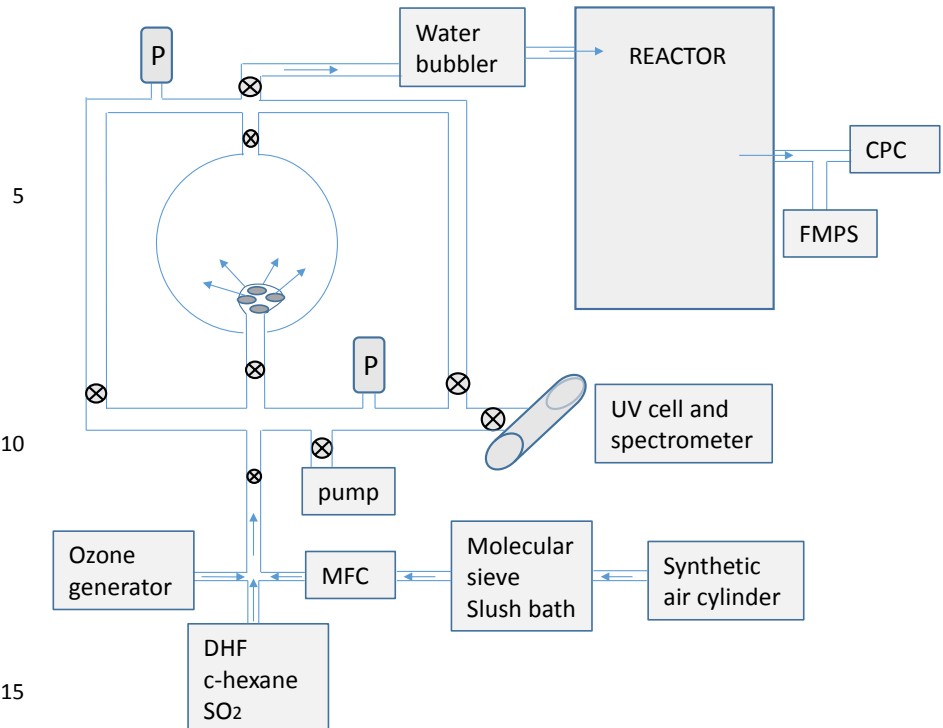

**Figure 1.** Schematic set-up of the experimental system.

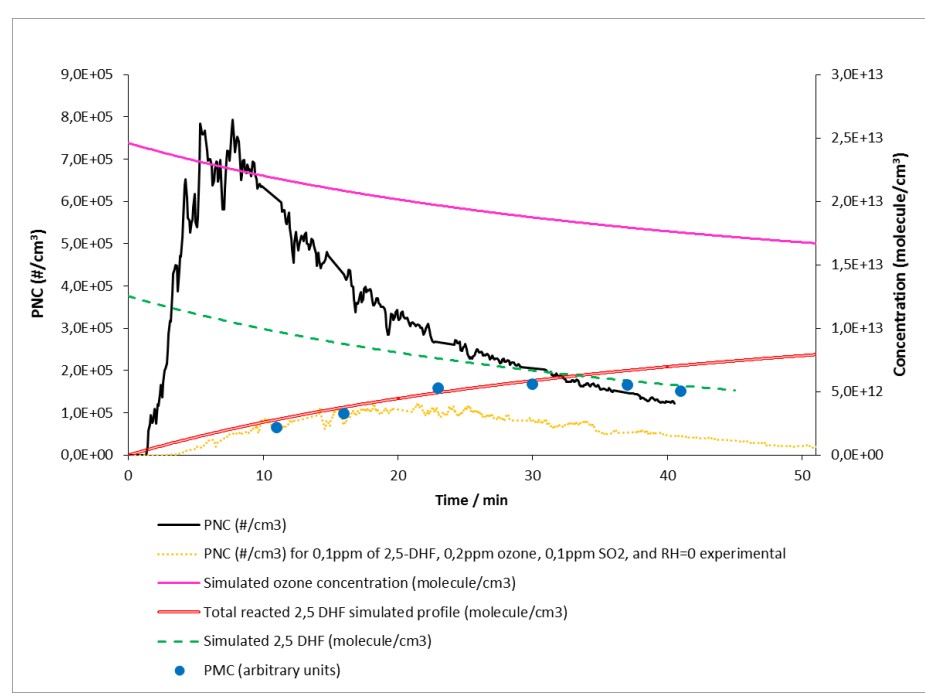





**Figure 2.** Temporal profiles for particles and gases. Initial concentrations: 0.5ppm of 2,5-DHF, 1ppm of $O_3$, 0.5ppm of $SO_2$ and RH= 0.

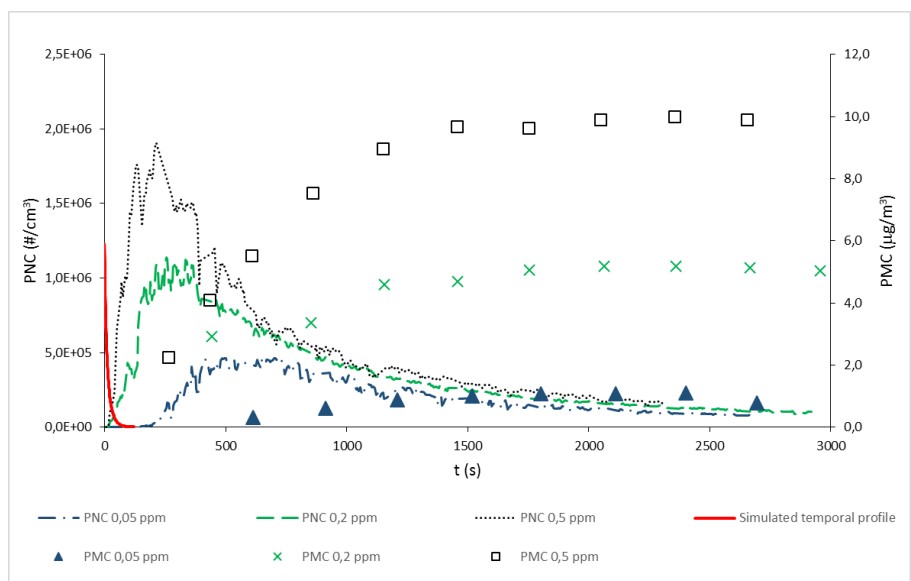

5    **Figure 3.** PNC and PMC profiles for different $SO_2$ initial concentrations ( 0.05, 0.2 and 0.5 ppm). This series of experiments was carried out under dry conditions and with 0.5 ppm of 2,3-DHF and 1 ppm of ozone initial concentrations. Simulated temporal profile of 2,3-DHF, red solid line.

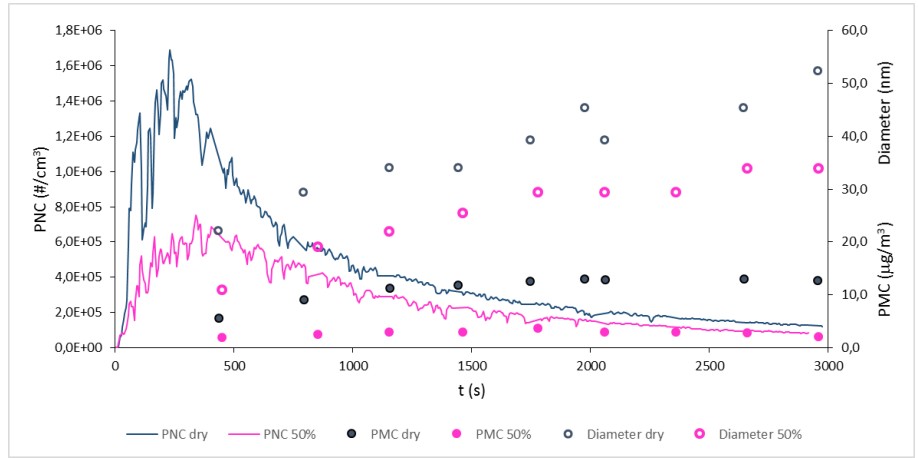

10    **Figure 4.** Effect of water vapour on PNC, PMC and on the particle diameters' profiles: dry conditions versus 50% HR. Both experiments were carried out with 0.5, 1.0 and 0.5 ppm initial concentrations of 2,3-DHF, ozone and $SO_2$ respectively.




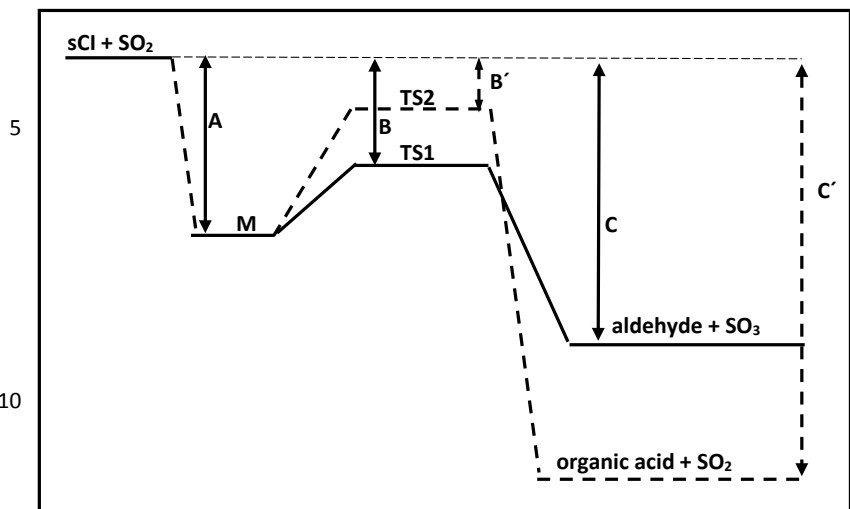

**Figure 5a**. General schematic potential energy surface for the reactions sCI + SO$_2$ → Products.



**Figure 5b.** Geometries of stationary points involved in the sCI + SO₂ reaction at BMK/6-311++G(3df,3pd) level of theory.