# Peer review of "Formation of secondary organic aerosols from the ozonolysis of dihydrofurans."

_Atmospheric Chemistry and Physics, 2016_

## Referee Comment (RC1) · Anonymous Referee #1 · 11 Nov 2016

Formation of secondary organic aerosols from the ozonolysis of dihydrofurans. Diaz de Mera, et al.

The manuscript's focus is to show nucleation of particles from the ozonolysis of 2,3-dihydrofuran and 2,5-dihydrofuran. Production of condensable gases is suggested to involve formation of organic acids from Criegee intermediates (CI) via catalysis by SO2. This mechanism is mainly supported by the fact that increasing water vapor reduced the observed nucleation. The authors suggest that higher water vapor concentrations compete for reaction with CI, reducing the fraction of CI that reacts with SO2. The authors' main argument is that CI are available to react with SO2 via a new mechanism that does not involve oxidation of SO2 to form sulfuric acid in the presence of water vapor. Notably, the proposed mechanisms appear to have an intermediate that involves SO3 (TS1.1, TS2.1, TS3.1). While it is stated that SO2 is not depleted, no data is

shown to support this statement. Previously SO2 has been shown to be oxidized by a number of CI, derived from different precursors, whether di-iodo species or from ozonolysis of alkenes.[1–5] To propose a new mechanism of reaction requires clearer evidence, particularly when measuring nucleation. Direct measurements of SO2 must be presented and it must be shown that the mass lost to particle formation would be easily detectable and above signal to noise of the SO2 detector. Although efforts were taken to remove water from the system and particle nucleation was still observed, low levels of residual water, from chamber walls perhaps, could provide water vapor. No direct measurement of humidity or water vapor was presented. Table 1 does not indicate which experiments included water.

The explanation of reaction of stabilized Criegee intermediates (SCI) with SO2 is problematic because the alkene reacted is small and cyclic, making it inherently unstable. Existing studies suggest that for such a small CI, stabilization will be negligible.[3,6–9] The energy released from the ozonolysis reaction will be in the range of 45 kCal/mol, all of this energy will remain in the resulting CI. Unimolecular decomposition should be the dominant pathway for these compounds. Studies showing reaction with SO2 used either a different route to SCI formation (di-iodo photolysis) or in fact detect oxidation of SO2. A seven member cyclic alkene, larger than the dihydrofurans by 2 carbons, showed yields of organic acids that were not strongly dependent on RH and were very low, less than a few percent.[10] While the proposed mechanism may occur, more information on its feasibility, in terms of the unimolecular reactions of the CI must be addressed. The energy of the transition state en route to the primary ozonide, or at least the primary ozonide itself, which indicate the reaction exothermicity, must be considered. Formation of SOA from oxidation is a complicated, multiphase process that is yet more complex due to deposition of condensable vapors and particles to the chamber walls, particularly for a the reactor used here, which has a low surface area to volume ratio.[11–13] For these reasons, inferring rate constants, even a ratio of rate constants, for the reactions leading to SOA formation is not reasonable without having any gas phase measurements. Data is not available for the decay of the furan or the

increase in condensable products. It is agreed that you observe increased humidity decreases SOA formation, but it data does not clearly explain the origin of this effect. Extrapolation of SOA formation data to rate constants of the oxidation reactions formation condensable products is not warranted. Those measurements are difficult enough to make, even when directly measuring the gas phase, without the complications of partition, both to particles and the chamber walls. If this analysis is to be used, more rigorous modeling of SOA formation and wall loss must be included.

The authors present clear indications of an interesting process leading to SOA formation from ozonolysis of compounds that have not had much study, but require somewhat speculative explanations in terms of the mechanism, particularly because the presented mechanism is at odds with existing knowledge of ozonolysis of small, cyclic alkenes. The observations and explanations may well be fully valid, but sufficient evidence, particularly concentrations of SO2 and some composition information on either the gas or particle phase, to support the mechanistic claims is not presented. Major revisions are required, including presentation of the SO2 concentrations and some rationalization of the formation of SCI from this dihydrofurans.

References

(1) Welz, O.; Savee, J. D.; Osborn, D. L.; Vasu, S. S.; Percival, C. J.; Shallcross, D. E.; Taatjes, C. a. Direct kinetic measurements of Criegee intermediate (CHâĆĆOO) formed by reaction of CHâĆĆI with OâĆĆ. Science 2012, 335 (6065), 204–207. (2) Lee, Y.-P. Perspective: Spectroscopy and kinetics of small gaseous Criegee intermediates. J. Chem. Phys. 2015, 143 (2), 20901. (3) Johnson, D.; Marston, G. The gas-phase ozonolysis of unsaturated volatile organic compounds in the tropospherew. 2008, 699–716. (4) Hatakeyama, S.; Kobayashi, H.; Akimoto, H. Gas-phase oxidation of sulfur dioxide in the ozone-olefin reactions. J. Phys. Chem. 1984, 88 (20), 4736–4739. (5) Osborn, D. L.; Taatjes, C. A. The physical chemistry of Criegee intermediates in the gas phase. Int. Rev. Phys. Chem. 2015, 34 (3), 309–360. (6) Donahue, N. M.; Drozd, G. T.; Epstein, S. a; Presto, A. a; Kroll, J. H. Adventures in ozoneland: down

the rabbit-hole. Phys. Chem. Chem. Phys. 2011, 13 (23), 10848–10857. (7) Drozd, G. T.; Donahue, N. M. Pressure dependence of stabilized Criegee intermediate formation from a sequence of alkenes. J. Phys. Chem. A 2011, 115 (17), 4381–4387. (8) Hakala, J. P.; Donahue, N. M. Pressure-Dependent Criegee Intermediate Stabilization from Alkene Ozonolysis. J. Phys. Chem. A 2016, 120 (14), 2173–2178. (9) Chuong, B.; Zhang, J.; Donahue, N. M. Cycloalkene Ozonolysis: Collisionally Mediated Mechanistic Branching. J. Am. Chem. Soc. 2004, 126 (39), 12363–12373. (10) Orzechowska, G. E.; Nguyen, H. T.; Paulson, S. E. Photochemical Sources of Organic Acids. 2. Formation of C5âĔȨ'C9 Carboxylic Acids from Alkene Ozonolysis under Dry and Humid Conditions. J. Phys. Chem. A 2005, 109 (24), 5366–5375. (11) Matsunaga, A.; Ziemann, P. J. Gas-Wall Partitioning of Organic Compounds in a Teflon Film Chamber and Potential Effects on Reaction Product and Aerosol Yield Measurements. Aerosol Sci. Technol. 2010, 44 (10), 881–892. (12) Zhang, X.; Cappa, C. D.; Jathar, S. H.; McVay, R. C.; Ensberg, J. J.; Kleeman, M. J.; Seinfeld, J. H.; Christopher D. Cappa. Influence of vapor wall loss in laboratory chambers on yields of secondary organic aerosol. Proc. Natl. Acad. Sci. U. S. A. 2014, 111 (16), 1–6. (13) La, Y. S.; Camredon, M.; Ziemann, P. J.; Valorso, R.; Matsunaga, A.; Lannuque, V.; Lee-Taylor, J.; Hodzic, A.; Madronich, S.; Aumont, B. Impact of chamber wall loss of gaseous organic compounds on secondary organic aerosol formation: Explicit modeling of SOA formation from alkane and alkene oxidation. Atmos. Chem. Phys. 2016, 16 (3), 1417–1431.

---

## Referee Comment (RC2) · Anonymous Referee #2 · 17 Nov 2016

This paper represents a new chemical mechanism potentially relevant for NPF, but the experimental data does not fully prove the proposed mechanism. Lack of experimental support is compensated by theoretical arguments and I guess this paper could be published once the problems pointed out by referee #1 and my concerns below are addressed.

Reviewer #1 already provided an extensive and complete review and I avoid repeating that but I still want to express my major concerns despite the overlap. My main concern is that authors quite vaguely exclude the SO3 channel and the role of sulphuric acid in observed NPF. Proper exclusion of SO3 channel is critical since authors are providing a new chemical mechanism arising from SO3 exclusion and explaining NPF in their system with previously unrecognised pathways. I cannot judge myself, if the proposed mechanism with SO2 catalyzing Criegee conversion to organic acids or aldehydes is

[Figure]

relevant or not, but I do question a) the absence of sulphuric acid in the system and b) the atmospheric relevance of the proposed mechanism even if correct and relevant in chamber conditions.

Authors state that reaction of SO3 with water vapour cannot be related to NPF since there is no water. Water is omnipresent even in authors' chamber and the reaction of SO3 with water is fast and unlikely seriously limited by the availability of water even in "dry" conditions. Water can come from the walls, and even through the Teflon wall, with trace gases and from the synthetic air bottle. Even if the lack of water would slow down the SO3 conversion to sulphuric acid, vast amounts of SO3 can be produced from sCI+SO2 or OH+SO2 and minute water concentrations could be enough for sufficient sulphuric acid formation. What are the yields of OH from ozonolysis of these DHFs?

OH scavenging was >95%. Still, with very high concentrations of furan and ozone, the OH production could potentially be high enough that the residual <5% reacting with SO2 (High concentrations up to 0.5ppm!) can be a significant source of SO3 and subsequently sulphuric acid under assumption that water residuals are present. More is required to show that not enough sulphuric acid can be formed via sCI+SO2/OH+SO2 -> SO3 (+H2O) -> H2SO4. (Even though the presented theoretical analyzis suggests SO3 is not released from the reaction of sCI and SO2, previous literature shows that's unlikely the case with most alkenes). With some assessment of water vapour concentration upper limit and with known or approximated reaction rates and yields authors could maybe get at least an idea about maximum sulphuric acid concentrations in the chamber.

The statement that SO2 remained constant (p6., l7.) is not supported by data shown. Was it measured? And if it was measured, with 0.5 ppm SO2, there's 1e13 molecules of it in a cm-3. If one per-mille of that is converted to H2SO4, that would be sufficient (concentration up to some 1e10 molec cm-3) to drive NPF with unavoidable background contaminants (e.g. ammonia or amines) or with some products from DHF oxidation and produce the observed NPF rate of maybe 1000 #/s. And a 1 per-mil drop

in SO2 may be tricky to observe. So what is really the experimental evidence for the SO2 recycling?

All in all, before publishing this paper, I would like to see more results and discussion to exclude the SO3 channel and sulphuric acid produced via that channel either purely from OH or with an assumption that theoretical prediction of no SO3 formation from sCI was incorrect. If, from experimental data, the presence of sulphuric acid cannot be excluded, the paper should be written in a manner that accounts for that deficiency.

I also have some doubts that any organic acids formed from these relatively small alkenes would have sufficiently low vapour pressures that they could homogeneously nucleate. (There are also methods to estimate the vapour pressure of such compounds, see e.g. Donahue et al. Atmos Chem Phys 11, 3303–3318, 2011; Pankow et al., Atmos Chem Phys 8, 2773–2796, 2008). These acids may be partitioning between particle/wall and gas phase, but that they underwent homogeneous nucleation with high nucleation rates is more questionable. If authors suggest nucleation is driven by proposed compounds, some more data or discussion would be appreciated.

On the other hand, the concentrations in the chamber system are vastly above the atmospheric ones (ozone, DHF and SO2 are 100-100 times higher than typical for the atmosphere), meaning that at least the atmospheric relevance of such acids, even if they were nucleating in the chamber, is more than questionable and against the present understanding on atmospheric nucleation processes.

Intro: Authors largely exclude the discussion related to extremely low volatile organics formed in auto-oxidation reactions from alkene ozonolysis (Ehn et al., Nature, 506, 476-, Nature; and many subsequent publications) considered to be one of the main pathways to atmospheric NPF and SOA. Since the title and motivation of this paper is SOA, the major progress on that field should be shortly discussed in the introduction.

---

## Referee Comment (RC3) · Anonymous Referee #3 · 21 Nov 2016

The authors describe experimental findings from the ozonolysis reaction of 2,3- and 2,5-dihydrofuran at atmospheric pressure and room temperature. Experiments were carried out in a Teflon bag with special attention to SOA formation. Particle formation was followed by total number measurements using a TSI CPC 3775 as well by measuring the particle size distribution by means of a TSI FMPS 3091. Gas-phase species, such as ozone, the dihydrofurans and water vapor, were not monitored in the course of the reaction. For runs in presence of SO2, the SO2 time series were only monitored "For some experiments" but no information on that is presented in the manuscript.

The authors concluded as a result of their experiments that i) the detected particle formation was not connected to sulfuric acid, ii) the reaction of Criegee intermediates (CI) with SO2 proceeds without SO3 formation and SO2 serves as a "catalyst" for acid formation starting from CI. These are very interesting statements. But unfortunately,

the experiments do not distinctly support the conclusions. A significant fraction of other products than sulfuric acid from CI+SO2 would be very important for the understanding of CI′s role in atmospheric oxidation.

Here my critical points:

- The authors used very high reactant concentrations, far away from atmospheric conditions. Why they are doing so? High initial concentrations connected with high intermediate concentrations can open reaction channels not relevant for the atmosphere. Particle measurements are sensitive enough allowing to work close to atmospheric reactant conditions.

- Cyclohexane was used as OH scavenger for more than 95% of the OH radicals. But what about the residual OH radicals? They are definitely reacting with SO2 in competition with all other OH reactions in the system forming finally sulfuric acid.

- What does it mean "completely dry conditions"? A measurement of the water vapor concentration in the Teflon bag is needed. It′s very challenging to produce and handle an extremely dry reaction gas with a water vapor concentration as low as needed that hydrolysis of SO3 doesn′t work!

- Sulfuric acid measurements are needed in order to rule out a significant contribution of sulfuric acid for nucleation and particle growth. I would encourage the authors to collaborate with groups familiar with the H2SO4 CIMS technique.

- The authors should show the SO2 time series or at least an example. What′s the uncertainty of the SO2 detection? I guess, from a constant SO2 time series of 10(12) molecules cm(-3) or more it is impossible concluding that 10(9) or 10(10) molecules cm(-3) have been converted. And that′s enough to produce the needed sulfuric acid for nucleation and early growth.

All together, I guess this manuscript needs a major revision based on additional experiments. It could become an important paper in the field of CI reactions. From my

perspective, at the moment the experimental basis is not good enough.

---

## Author Comment (AC1) · 13 Dec 2016

Answer to referee#1. We sincerely thank the comments of the referee. Additional information is now included together with data from new experiments carried out during these few weeks in line with the requirements of the referee. The mechanism for sCI reaction with SO2 to generate organic acids and SO2 is not really new. Other studies have found the formation of an acid and the release of SO2 from the secondary ozonide as the most energetically favourable reaction channel for even smaller sCI, (Kurten, 2011; Vereecken, 2012). The experimental results obtained in this study support those previous theoretical works. "Table 1 does not indicate which experiments include water": All the experiment summarised in Table 1 were carried out with HR=0. It was stated in page 7, line 10. For sake of clarity now this information is included also in the table in the revised manuscript. As suggested by the referee, the energy

of the reactants (alkene + SO2) and the primary ozonide have been included in table 2 and in figure 5. Due to the exothermic nature of the ozonolysis, unimolecular decomposition and stabilisation through collision would compete in the formation of sCI. A comment in this sense has been included in the manuscript. Concerning to water concentrations, we have RH meters but their ranges are not useful for RH below 1%. The stated water ratio from the cylinder is 1-2ppm (Praxair). We have tried to calculate the water concentration in the synthetic air after drying it through a LN2 trap with molecular sieve 5A using a mass spectrometer but the residual H2O signal from chamber degas is too high to enable quantification. We have also tried with a FTIR but the purge instability in the FTIR was above the expected changes in the water signal. From the literature-available data for ice, passing the synthetic air through the LN2 trap, the vapour pressure is expected to fall well below 10-7Pa (Murphy and Koop, 2005) and thus the expected concentration in the dried synthetic air would be below 1x107molecule cm-3. Since residual concentrations could be higher, we have conducted a series of experiments to test the possibility of SO3–water reaction in the reactor and to estimate the water concentration. We have used a degassed sample of solid sulfur trioxide (99.5%, stabilized, Aldrich) contained in a glass flask to obtain different SO3 concentrations in the reactor (as it was done in previous studies (Jayne, 1997). Freshly dried synthetic air was mixed with SO3 in the teflon reactor and the mixture was continuously monitored by the CPC for 40 minutes. The figure shows the results for experiments with SO3 concentration in the range 1 to 12ppb. No particles could be observed for experiments with low initial SO3 concentrations. On the other hand, NPF was observed for the experiments with SO3 in the range 6 to 12 ppb. Under these experimental conditions, nucleation is attributed to the formation of H2SO4 from the reaction of SO3 and the residual H2O. The overall gas-phase reaction H2O + SO3 → H2SO4 exhibits a second-order dependence on water vapor concentration, the first-order rate coefficient for the SO3 loss being k= 3.90x10-41exp(6830.6/T)[H2O]2 (Jayne, 1997). Taking into account that the approximate H2SO4 gas phase concentration able to nucleate is around 5x106molecule cm-3 (Metzger, 2010), the concentration

of water in the reactor may be obtained by simulating the SO3 and H2SO4 profiles for different guessed H2O profiles. Thus for example, since no NPF was observed for the experiment with 2ppb of SO3, the water concentration must be below 15ppb. On the other hand from the experiment with 6ppb of SO3, a 20 ppb water concentration is required to reproduce the observed nucleation. From all the experiments carried out, we estimate that water concentration in the reactor is 20ïĆś10ppb. To check for permeation through the reactor wall, some experiments have been also carried out with dry air after 1 hour in the reactor. The results were similar to those carried out with freshly dried air.

Figure 1

This figure and the related discussion has been included in the supporting information.

According to the SO2 comments, direct measurements of SO2 are included in this reply and in the revised manuscript. Furthermore, these weeks we had an ozone analyser available that provided additional information concerning the gas phase (Figure 2). The experimental and simulated profiles of ozone (based on the literature ozonolysis rate constant) are in good agreement and so ozone is not quantitatively lost through other reactions. Figure 2

We have carried out new experiments with lower SO2 concentrations in the range 10-20ppb. In all cases the profile of SO2 remained neatly constant. For example it was 10.0ïĆś0.2 ppb during the whole experiment for the experiments with 10ppb of SO2. A first order loss rate constant may be derived from the SO2 profile: $k= 8.5 \times 10-6 s-1$. Thus, SO2 losses (if they occur) are very low. See the figure.

Figure 3 To check the possibility of SO3 production, we can assume a simple mechanism where any lost SO2 molecule would be converted exclusively into SO3: SO2 $\rightarrow$ SO3 $k= 8.5 \times 10-6 s-1$ [SO2]o = 10ppb And then SO3 would exclusively react with water to produce H2SO4: H2O + SO3 $\rightarrow$ H2SO4 $k= 3.90 \times 10-41 exp(6830.6/T)[H2O]2$ (Jayne, 1997)

[Figure]

Simulating the SO2, SO3 and H2O profiles for a 20ppb water concentration and for 10ppb initial SO2 concentration, it would require more than 1 hour to generate 5x106molecule cm-3 of H2SO4, which is the approximate concentration able to nucleate (Metzger, 2010). For 20ppb initial SO2 concentration it would require 28 minutes. Nevertheless, for these experiment nucleation was visible at 2minutes (almost instantaneous if we take away the mixing time of reactants). Thus, considering the low levels of water vapour in the reactor and the observation of nearly constant SO2 for the experiments with lower SO2 concentrations, the contribution of SO3-H2SO4 pathway to NPF seems to be minor and unable to lead to nucleation by itself. In this sense the catalytic pathway releasing SO2, which is thermodynamic- favourable, may be the key to NPF.

Main changes in the manuscript. Page 1, line 14. It has been rewritten. Water presence at ppb-ppm concentration may have an effect on SOA production. Nevertheless, for higher concentrations, no effect was found. Page 1, line 18. SO3 role is not overall ruled out. SO2 catalysed reactions are suggested as an additional pathway to NPF. Page 1, line 18. SO3 is not excluded as a possible intermediate to produce SOA. Page 3, line 31. The term "completely dry conditions" has been removed. Page 4, line 4. The estimated water concentration inside the reactor is stated. Page 4, line 22. The experiments carried out with the ozone analyser are introduced. Page 6. Line 9. New data and discussion about SO2, SO3 and sulfuric acid is introduced. Page 7, line 21. New data and discussion about SO2, SO3 and sulfuric acid is introduced. Page 9, line 29. The energy of the first step of the ozonolysis is introduced. Page 10, line 18. From the results of this study, (R4) is suggested as the probable pathway to NPF. Page 10, line 24. Vapour pressure estimates are given. Page 11, line 14. SO3 role is not overall ruled out. The statements concerning SO2 are restricted to low SO2 concentration conditions. Table 2. The optimized energies of the reactants and first ozonides are included in the table. Figure 2 includes experimental gas-phase profiles for SO2 and O3. Figure 3 includes the O3 experimental profile. Figure 5a. Reactants and the first ozonides have been included in the mechanism scheme. New figures, S1 and S3,

have been introduced in the supporting information.

References: Jayne, J. T., Po1schl, U., Chen, Y., Dai, D., Molina, L.T., Worsnop, D.R., Kolb, C.E., Molina, M. J.: Pressure and Temperature Dependence of the Gas-Phase Reaction of SO3 with H2O and the Heterogeneous Reaction of SO3 with H2O/H2SO4 Surfaces. J. Phys. Chem. A, 101, 10000-10011, 1997. Kurtén, T., Lane, J. R., Jørgensen, S., Kjaergaard, H. G.: A Computational Study of the Oxidation of SO2 to SO3 by Gas-Phase Organic Oxidants. J. Phys. Chem. A,116, 6823-6830, 2011. Metzger, A., Verheggen, B., Dommen, J., Duplissy, J., Prevota, A. S. H., Weingartner, E.,Riipinen, I, Kulmala, M., Spracklend, D. V., Carslaw, K. S., Baltensperger, U.: Evidence for the role of organics in aerosol particle formation under atmospheric conditions. PNAS. 107, 6646-6651, 2010. Murphy, D., M. and Koop, T.: Review of the vapour pressures of ice and supercooled water for atmospheric applications, Quart. J. Royal Meterol. Soc., 131, 1539–1565, 2005. Vereecken, L., Harder, H., Novelli, A.: The reaction of Criegee intermediates with NO, RO2, and SO2, and their fate in the atmosphere, Phys. Chem. Chem. Phys., 14, 14682–14695, 2012.

Please also note the supplement to this comment:
http://www.atmos-chem-phys-discuss.net/acp-2016-891/acp-2016-891-AC1-supplement.pdf
* * *
[Figure]

[Figure]

[Figure]

**Fig. 2.** Figure 2

[Figure]

**Fig. 3.** Figure 3

---

## Author Comment (AC2) · 13 Dec 2016

Answer to referee #2. We sincerely thank the comments of the referee. Additional information is now included together with data from new experiments carried out during these few weeks in line with the requirements of the referee. The mechanism for sCI reaction with SO2 to generate organic acids and SO2 is not really new. Other studies have found the formation of an acid and the release of SO2 from the secondary ozonide as the most energetically favourable reaction channel for even smaller sCI (Jiang, 2010; Kurten, 2011; Vereecken, 2012). The experimental results obtained in this study are consistent with the isomerisation channel proposed in these previous theoretical works. a)"Absence of sulphuric acid in the system". From the literature-available data for ice, passing the synthetic air through the LN2 trap, the vapour pressure is expected to fall well below 10-7Pa (Murphy and Koop, 2005) and thus the expected concentration in

the dried synthetic air is below 1x107molecule cm-3. Since residual concentrations could be higher, we have conducted a series of experiments to test the possibility of SO3–water reaction in the reactor and to estimate the water concentration. We have used a degassed sample of solid sulfur trioxide (99.5%, stabilized, Aldrich) contained in a glass flask to obtain different SO3 concentrations in the reactor (as it was done in previous studies (Jayne, 1997). Freshly dried synthetic air was mixed with SO3 in the teflon reactor and the mixture was continuously monitored by the CPC for 40 minutes. The figure shows the results for experiments with SO3 concentration in the range 1 to 12ppb. No particles could be observed for experiments with low initial SO3 concentrations. On the other hand, NPF was observed for the experiments with SO3 in the range 6 to 12 ppb. Under these experimental conditions, nucleation is attributed to the formation of H2SO4 from the reaction of SO3 and the residual H2O. The overall gas-phase reaction H2O + SO3 → H2SO4 exhibits a second-order dependence on water vapor concentration, the first-order rate coefficient for the SO3 loss being k= 3.90x10-41exp(6830.6/T)[H2O]2 (Jayne, 1997). Taking into account that the approximate H2SO4 gas phase concentration able to nucleate is around 5x106molecule cm-3 (Metzger, 2010), the concentration of water in the reactor may be obtained by simulating the SO3 and H2SO4 profiles for different guessed H2O profiles. Thus for example, since no NPF was observed for the experiment with 2ppb of SO3, the water concentration must be below 15ppb. On the other hand from the experiment with 6ppb of SO3, a 20ppb water concentration is required to reproduce the observed nucleation. From all the experiments carried out, we estimate that water concentration in the reactor is 20ïĆś10ppb. To check for permeation through the reactor wall, some experiments have been also carried out with dry air after 1 hour in the reactor. The results were similar to those carried out with freshly dried air. Figure 1

This figure and the related discussion has been included in the supporting information.

According to the SO2 comments, direct measurements of SO2 are included in this reply and in the revised manuscript. We have carried out new experiments with lower

SO2 concentrations in the range 10-20ppb. In all cases the profile of SO2 remained neatly constant. For example it was 10.0+-0.2ppb during the whole experiment for the experiments with 10ppb of SO2. A first order loss rate constant may be derived from the SO2 profile: k= $8.5 \times 10^{-6} s^{-1}$. Thus, SO2 losses (if they occur) are very low. See the figure. Figure 2

To check the possibility of SO3 production, we can assume a simple mechanism where any lost SO2 molecule would be converted exclusively into SO3: $SO2 \rightarrow SO3$ k= $8.5 \times 10^{-6} s^{-1}$ $[SO2]_o = 10ppb$ And then SO3 would exclusively react with water to produce H2SO4: $H2O + SO3 \rightarrow H2SO4$ k= $3.90 \times 10^{-41} exp(6830.6/T)[H2O]^2$ (Jayne, 1997)

Simulating the SO2, SO3 and H2O profiles for a 20ppb water concentration and for 10ppb initial SO2 concentration, it would require more than 1 hour to generate $5 \times 10^6$ molecule $cm^{-3}$ of H2SO4, which is the approximate concentration able to nucleate (Metzger, 2010). For 20ppb initial SO2 concentration it would require 28 minutes. Nevertheless, for these experiment nucleation was visible at 2minutes (almost instantaneous if we take away the mixing time of reactants). Relatively high levels of OH would deplete SO2 if SO2 concentration were very low. In this sense, the experiments with high dihydrofurans and ozone concentrations (for example 0.5 and 1.0 ppm, respectively) and low SO2 concentration (in the range of 10ppb) show that residual OH concentration must be negligible in the system since the experimental SO2 concentration remained constant during the experiments. Thus, considering the low levels of water vapour in the reactor and the observation of nearly constant SO2 for the experiments with lower SO2 concentrations, the contribution of SO3-H2SO4 pathway to NPF seems to be minor and unable to lead to nucleation by itself. In this sense the catalytic pathway releasing SO2, which is thermodynamic-favourable, may be the key to NPF. For higher SO2 concentrations (in the range of 0.5 ppm) small changes at the level of the uncertainty of the SO2 measurements can not be completely excluded as a possible source of SO3.

b)Atmospheric relevance. We have carried out experiments at lower concentrations. Thus, for example, for 2,5-DHF no particles were detected for 0.02, 0.04 and 0.02ppm concentrations of 2,5-DHF, ozone and SO2, respectively. For 0.05, 0.1 and 0.05 ppm concentrations (2 ,5-DHF, ozone and SO2) NPF could be observed but particle number concentration, particle size diameter and particle mass concentration were very low and noisy and could not be measured accurately. To assess the effects of water, SO2 and ozone, the concentrations had to be increased. Although the concentrations of reactants in this study are higher than average concentrations in the atmosphere and nucleation from the ozonolysis of only DHFs is not expected, this work shows that these reaction lead to condensable species that could contribute to NPF or particle growing in the atmosphere. Furthermore this study reports theoretical and experimental data that points the catalytic role of SO2 in the oxidation of SCIs.

Vapour pressure. From the work of Donahue et al (2011) a saturation mass concentration around 100 microgram.m-3 is expected for C4 chemicals with a 1:1 oxygenation ratio (O:C). It is the high degree of oxygenation which leads to a low volatility value, 5x1011molecules cm-3. For the range of initial reactant concentrations in the laboratory experiments this range of product concentration could be reached. A comment has been introduced in the manuscript as suggested by the referee. Ehn 2014 reference has been included in the introduction as suggested by the referee.

Main changes in the manuscript. Page 1, line 14. It has been rewritten. Water presence at ppb-ppm concentration may have an effect on SOA production. Nevertheless, for higher concentrations, no effect was found. Page 1, line 18. SO3 role is not overall ruled out. SO2 catalysed reactions are suggested as an additional pathway to NPF. Page 1, line 18. SO3 is not excluded as a possible intermediate to produce SOA. Page 3, line 31. The term "completely dry conditions" has been removed. Page 4, line 4. The estimated water concentration inside the reactor is stated. Page 4, line 22. The experiments carried out with the ozone analyser are introduced. Page 6. Line 9. New data and discussion about SO2, SO3 and sulfuric acid is introduced. Page 7, line 21.

New data and discussion about SO2, SO3 and sulfuric acid is introduced. Page 9, line 29. The energy of the first step of the ozonolysis is introduced. Page 10, line 18. From the results of this study, (R4) is suggested as the probable pathway to NPF. Page 10, line 24. Vapour pressure estimates are given. Page 11, line 14. SO3 role is not overall ruled out. The statements concerning SO2 are restricted to low SO2 concentration conditions. Table 2. The optimized energies of the reactants and first ozonides are included in the table. Figure 2 includes experimental gas-phase profiles for SO2 and O3. Figure 3 includes the O3 experimental profile. Figure 5a. Reactants and the first ozonides have been included in the mechanism scheme. New figures, S1 and S3, have been introduced in the supporting information.

References: Jayne, J. T., Po1schl, U., Chen, Y., Dai, D., Molina, L.T., Worsnop, D.R., Kolb, C.E., Molina, M. J.: Pressure and Temperature Dependence of the Gas-Phase Reaction of SO3 with H2O and the Heterogeneous Reaction of SO3 with H2O/H2SO4 Surfaces. J. Phys. Chem. A, 101, 10000-10011, 1997. Kurtén, T., Lane, J. R., Jørgensen, S., Kjaergaard, H. G.: A Computational Study of the Oxidation of SO2 to SO3 by Gas-Phase Organic Oxidants. J. Phys. Chem. A,116, 6823-6830, 2011. Metzger, A., Verheggen, B., Dommen, J., Duplissy, J., Prevota, A. S. H., Weingartner, E.,Riipinen, I, Kulmala, M., Spracklend, D. V., Carslaw, K. S., Baltensperger, U.: Evidence for the role of organics in aerosol particle formation under atmospheric conditions. PNAS. 107, 6646-6651, 2010. Murphy, D., M. and Koop, T.: Review of the vapour pressures of ice and supercooled water for atmospheric applications, Quart. J. Royal Meterol. Soc., 131, 1539–1565, 2005. Vereecken, L., Harder, H., Novelli, A.: The reaction of Criegee intermediates with NO, RO2, and SO2, and their fate in the atmosphere, Phys. Chem. Chem. Phys., 14, 14682–14695, 2012.

Please also note the supplement to this comment:
http://www.atmos-chem-phys-discuss.net/acp-2016-891/acp-2016-891-AC2-supplement.pdf

[Figure]

Chart:
- Left Y-axis: PNC (#/cm³), from 0,0E+00 to 3,5E+04
- Right Y-axis: H₂SO₄ simulated concentration (molecule/cm³), from 0,0E+00 to 1,0E+07
- X-axis: t (s), from 0 to 2000+

Legend:
- PNC for [SO3]=2 ppb
- PNC for [SO3]=6 ppb
- PNC for [SO3]=12 ppb
- [H2SO4] simulated for [SO3]=2 ppb and [H2O]=15 ppb
- [H2SO4] simulated for [SO3]=6 ppb and [H2O]=20 ppb
- [H2SO4] simulated for [SO3]=12 ppb and [H2O]=20 ppb

[Figure]

**Supplement:**

[revised manuscript text omitted]

---

## Author Comment (AC3) · 13 Dec 2016

Answer to referee #3. We sincerely thank the comments of the referee. Additional information is now included together with data from new experiments carried out during these few weeks in line with the requirements of the referee.

High reactant concentrations. We have carried out experiments at lower concentrations. Thus, for example, for 2,5-DHF no particles were detected for 0.02, 0.04 and 0.02 ppm concentrations of 2,5-DHF, ozone and SO2, respectively. For 0.05, 0.1 and 0.05 ppm concentrations (2 ,5-DHF, ozone and SO2) NPF could be observed but particle number concentration, particle size diameter and particle mass concentration were very low and noisy and could not be measured accurately. To assess the effects of water, SO2 and ozone, the concentrations had to be increased. "Completely dry

conditions" and sulphuric acid measurements. This term has been removed from the manuscript. For the experiments with dried synthetic air (passing through a liquid nitrogen trap), the concentration of water in the reactor has been estimated. We have conducted a series of experiments to test the possibility of SO3 –water reaction in the reactor and to estimate the water concentration. We have used a degassed sample of solid sulfur trioxide (99.5%, stabilized, Aldrich) contained in a glass flask to obtain different SO3 concentrations in the reactor (as it was done in previous studies (Jayne, 1997). Freshly dried synthetic air was mixed with SO3 in the teflon reactor and the mixture was continuously monitored by the CPC for 40 minutes. The figure shows the results for experiments with SO3 concentration in the range 1 to 12ppb. No particles could be observed for experiments with low initial SO3 concentrations. On the other hand, NPF was observed for the experiments with SO3 in the range 6 to 12 ppb. Under these experimental conditions, nucleation is attributed to the formation of H2SO4 from the reaction of SO3 and the residual H2O. The overall gas-phase reaction H2O + SO3 → H2SO4 exhibits a second-order dependence on water vapor concentration, the first-order rate coefficient for the SO3 loss being k= 3.90x10-41exp(6830.6/T)[H2O]2 (Jayne, 1997). Taking into account that the approximate H2SO4 gas phase concentration able to nucleate is around 5x106molecule cm-3 (Metzger, 2010), the concentration of water in the reactor may be obtained by simulating the SO3 and H2SO4 profiles for different guessed H2O profiles. Thus for example, since no NPF was observed for the experiment with 2ppb of SO3, the water concentration must be below 15ppb. On the other hand from the experiment with 6ppb of SO3, a 20ppb water concentration is required to reproduce the observed nucleation. From all the experiments carried out, we estimate that water concentration in the reactor is 20+-10ppb. To check for permeation through the reactor wall, some experiments have been also carried out with dry air after 1 hour in the reactor. The results were similar to those carried out with freshly dried air.

Figure 1

This figure and the related discussion has been included in the supporting information.

SO2 time series. According to the SO2 comments, direct measurements of SO2 are included in this reply and in the revised manuscript. We have also carried out new experiments with lower SO2 concentrations in the range 10-20ppb. In all cases the profile of SO2 remained neatly constant. For example it was $10.0\pm0.2$ ppb during the whole experiment for the experiments with 10ppb of SO2. A first order loss rate constant may be derived from the SO2 profile: k= 8.5x10-6s-1. Thus, SO2 losses (if they occur) are very low. See the figure. Figure 2

To check the possibility of SO3 production, we can assume a simple mechanism where any lost SO2 molecule would be converted exclusively into SO3: SO2 → SO3 k= 8.5x10-6s-1 [SO2]o = 10ppb And then SO3 would exclusively react with water to produce H2SO4: H2O + SO3 →H2SO4 k= 3.90x10-41exp(6830.6/T)[H2O]2 (Jayne, 1997) Simulating the SO2, SO3 and H2O profiles for a 20ppb water concentration and for 10ppb initial SO2 concentration, it would require more than 1 hour to generate 5x106molecule cm-3 of H2SO4, which is the approximate concentration able to nucleate (Metzger, 2010). For 20ppb initial SO2 concentration it would require 28 minutes. Nevertheless, for these experiment nucleation was visible at 2minutes (almost instantaneous if we take away the mixing time of reactants). Thus, considering the low levels of water vapour in the reactor and the observation of nearly constant SO2 for the experiments with lower SO2 concentrations, the contribution of SO3-H2SO4 pathway to NPF seems to be minor and unable to lead to nucleation by itself. In this sense the catalytic pathway releasing SO2, which is thermodynamic- favourable, may be the key to NPF.

Cyclohexane - OH excavenger. Relatively high levels of OH would deplete SO2 if SO2 concentration were very low. Nevertheless, even in those experiments where OH level could be higher (the experiments with dihydrofurans and ozone concentrations in the range of 0.5 and 1.0 ppm, respectively) and low SO2 concentration (in the range of 10ppb), the SO2 concentration did not fall during the experiment. These results suggest that that residual OH concentration must be negligible in the system.

Main changes in the manuscript. Page 1, line 14. It has been rewritten. Water presence at ppb-ppm concentration may have an effect on SOA production. Nevertheless, for higher concentrations, no effect was found. Page 1, line 18. SO3 role is not overall ruled out. SO2 catalysed reactions are suggested as an additional pathway to NPF.

Page 1, line 18. SO3 is not excluded as a possible intermediate to produce SOA.

Page 3, line 31. The term "completely dry conditions" has been removed.

Page 4, line 4. The estimated water concentration inside the reactor is stated.

Page 4, line 22. The experiments carried out with the ozone analyser are introduced.

Page 6. Line 9. New data and discussion about SO2, SO3 and sulfuric acid is introduced.

Page 7, line 21. New data and discussion about SO2, SO3 and sulfuric acid is introduced.

Page 9, line 29. The energy of the first step of the ozonolysis is introduced.

Page 10, line 18. From the results of this study, (R4) is suggested as the probable pathway to NPF.

Page 10, line 24. Vapour pressure estimates are given.

Page 11, line 14. SO3 role is not overall ruled out. The statements concerning SO2 are restricted to low SO2 concentration conditions.

Table 2. The optimized energies of the reactants and first ozonides are included in the table.

Figure 2 includes experimental gas-phase profiles for SO2 and O3.

Figure 3 includes the O3 experimental profile.

Figure 5a. Reactants and the first ozonides have been included in the mechanism

[Figure]

scheme.

New figures, S1 and S3, have been introduced in the supporting information.

References: Jayne, J. T., Po1schl, U., Chen, Y., Dai, D., Molina, L.T., Worsnop, D.R., Kolb, C.E., Molina, M. J.: Pressure and Temperature Dependence of the Gas-Phase Reaction of SO3 with H2O and the Heterogeneous Reaction of SO3 with H2O/H2SO4 Surfaces. J. Phys. Chem. A, 101, 10000-10011, 1997. Kurtén, T., Lane, J. R., Jørgensen, S., Kjaergaard, H. G.: A Computational Study of the Oxidation of SO2 to SO3 by Gas-Phase Organic Oxidants. J. Phys. Chem. A,116, 6823-6830, 2011. Metzger, A., Verheggen, B., Dommen, J., Duplissy, J., Prevota, A. S. H., Weingartner, E.,Riipinen, I, Kulmala, M., Spracklend, D. V., Carslaw, K. S., Baltensperger, U.: Evidence for the role of organics in aerosol particle formation under atmospheric conditions. PNAS. 107, 6646-6651, 2010. Murphy, D., M. and Koop, T.: Review of the vapour pressures of ice and supercooled water for atmospheric applications, Quart. J. Royal Meterol. Soc., 131, 1539–1565, 2005. Vereecken, L., Harder, H., Novelli, A.: The reaction of Criegee intermediates with NO, RO2, and SO2, and their fate in the atmosphere, Phys. Chem. Chem. Phys., 14, 14682–14695, 2012.

Please also note the supplement to this comment:
http://www.atmos-chem-phys-discuss.net/acp-2016-891/acp-2016-891-AC3-supplement.pdf

––––––––––––––––––––––––––––––––

[Figure]

[Figure]

**Supplement:**

[revised manuscript text omitted]

---

## Author Comment (AC5) · 13 Dec 2016

**Formation of secondary organic aerosols from the ozonolysis of dihydrofurans.**

Yolanda Diaz-de-Mera, Alfonso Aranda[*], Larisa Bracco, Diana Rodriguez and Ana Rodriguez.

**Supplementary Material.**

**Water vapour concentration for experiments with dried air samples.**

A series of experiments was conducted to estimate the water concentration under "dry conditions". We have used a degassed sample of solid sulfur trioxide (>99%, stabilized, Aldrich) contained in a glass flask to obtain different $SO_3$ concentrations in the reactor (as it was done in previous studies (Jayne, 1997). Freshly dried synthetic air from a liquid nitrogen trap was mixed with $SO_3$ in the teflon reactor and the mixture was continuously monitored by the CPC for 40 minutes. Fig. S1 shows the results for experiments with $SO_3$ concentration in the range 1 to 12ppb. No particles could be observed for experiments with low initial $SO_3$ concentrations. On the other hand, NPF (new particle formation) was observed for the experiments with $SO_3$ in the range 6 to 12 ppb. Under these experimental conditions, nucleation is attributed to the formation of $H_2SO_4$ from the reaction of $SO_3$ and the residual $H_2O$.

The overall gas-phase reaction $H_2O + SO_3 \rightarrow H_2SO_4$ exhibits a second-order dependence on water vapor concentration, the first-order rate coefficient for the $SO_3$ loss being k= $3.90 \times 10^{-41} \exp(6830.6/T)[H_2O]^2$ (Jayne, 1997)

Taking into account that the approximate $H_2SO_4$ gas phase concentration able to nucleate is around $5 \times 10^6$ molecule $cm^{-3}$ (Metzger, 2010), the concentration of water in the reactor may be obtained by simulating the $SO_3$ and $H_2SO_4$ profiles for different guessed $H_2O$ profiles. Thus for example, since no NPF was observed for the experiment with 2ppb of $SO_3$, the expected water concentration must be below 15ppb. On the other hand from the experiment with 6ppb of $SO_3$, a 20 ppb water concentration is required to reproduce the observed nucleation time. From all the experiments carried out, we estimate that the residual water concentration in the reactor is 20±10ppb. To check for permeation through the reactor wall, some experiments have been also carried out with dry air after 1 hour in the reactor. The results were similar to those carried out with freshly dried air.

[Figure]

**Figure S1**. Particle number concentration for different the reaction of $SO_3$ with residual $H_2O$ and simulated profiles of $H_2SO_4$ for given $SO_3$ and $H_2O$ concentrations.

[Figure]

**Figure S2.** Maximum particle number concentration and maximum mass concentration in experiments with different $SO_2$ initial concentrations. For this series of experiments, the initial concentrations of 2,5-DHF and ozone were 0.5 and 1.0 ppm respectively. Relative humidity=0.

[Figure]

**Figure S3.** $SO_2$ profiles during ozonolysis reactions of 2,5-DHF and ozone (0.5 and 1.0 initial concentrations, respectively) starting from different $SO_2$concentrations. Similar results were obtained for 2,3-DHF.

[revised manuscript text omitted]

---

## Author Response (AR1)

Formation of secondary organic aerosols from the ozonolysis of dihydrofurans. Diaz de Mera, et al.

The manuscript's focus is to show nucleation of particles from the ozonolysis of 2,3-dihydrofuran and 2,5-dihydrofuran. Production of condensable gases is suggested to involve formation of organic acids from Criegee intermediates (CI) via catalysis by $SO_2$. This mechanism is mainly supported by the fact that increasing water vapor reduced the observed nucleation. The authors suggest that higher water vapor concentrations compete for reaction with CI, reducing the fraction of CI that reacts with $SO_2$. The authors' main argument is that CI are available to react with $SO_2$ via a new mechanism that does not involve oxidation of $SO_2$ to form sulfuric acid in the presence of water vapor. Notably, the proposed mechanisms appear to have an intermediate that involves $SO_3$ (TS1.1, TS2.1, TS3.1). While it is stated that $SO_2$ is not depleted, no data is

shown to support this statement. Previously $SO_2$ has been shown to be oxidized by a number of CI, derived from different precursors, whether di-iodo species or from ozonolysis of alkenes.[1–5] To propose a new mechanism of reaction requires clearer evidence, particularly when measuring nucleation. Direct measurements of $SO_2$ must be presented and it must be shown that the mass lost to particle formation would be easily detectable and above signal to noise of the $SO_2$ detector. Although efforts were taken to remove water from the system and particle nucleation was still observed, low levels of residual water, from chamber walls perhaps, could provide water vapor. No direct measurement of humidity or water vapor was presented. Table 1 does not indicate which experiments included water.

The explanation of reaction of stabilized Criegee intermediates (SCI) with $SO_2$ is problematic because the alkene reacted is small and cyclic, making it inherently unstable. Existing studies suggest that for such a small CI, stabilization will be negligible.[3,6–9] The energy released from the ozonolysis reaction will be in the range of 45 kCal/mol, all of this energy will remain in the resulting CI. Unimolecular decomposition should be the dominant pathway for these compounds. Studies showing reaction with $SO_2$ used either a different route to SCI formation (di-iodo photolysis) or in fact detect oxidation of $SO_2$. A seven member cyclic alkene, larger than the dihydrofurans by 2 carbons, showed yields of organic acids that were not strongly dependent on RH and were very low, less than a few percent.[10] While the proposed mechanism may occur, more information on its feasibility, in terms of the unimolecular reactions of the CI must be addressed. The energy of the transition state en route to the primary ozonide, or at least the primary ozonide itself, which indicate the reaction exothermicity, must be considered. Formation of SOA from oxidation is a complicated, multiphase process that is yet more complex due to deposition of condensable vapors and particles to the chamber walls, particularly for a the reactor used here, which has a low surface area to volume ratio.[11–13] For these reasons, inferring rate constants, even a ratio of rate constants, for the reactions leading to SOA formation is not reasonable without having any gas phase measurements. Data is not available for the decay of the furan or the

increase in condensable products. It is agreed that you observe increased humidity decreases SOA formation, but it data does not clearly explain the origin of this effect. Extrapolation of SOA formation data to rate constants of the oxidation reactions formation condensable products is not warranted. Those measurements are difficult enough to make, even when directly measuring the gas phase, without the complications of partition, both to particles and the chamber walls. If this analysis is to be used, more rigorous modeling of SOA formation and wall loss must be included.

The authors present clear indications of an interesting process leading to SOA formation from ozonolysis of compounds that have not had much study, but require somewhat speculative explanations in terms of the mechanism, particularly because the presented mechanism is at odds with existing knowledge of ozonolysis of small, cyclic alkenes. The observations and explanations may well be fully valid, but sufficient evidence, particularly concentrations of $SO_2$ and some composition information on either the gas or particle phase, to support the mechanistic claims is not presented. Major revisions are required, including presentation of the $SO_2$ concentrations and some rationalization of the formation of SCI from this dihydrofurans.

**Answer to referee#1.**

We sincerely thank the comments of the referee. Additional information is now included together with data from new experiments carried out during these few weeks in line with the requirements of the referee.

The mechanism for sCI reaction with $SO_2$ to generate organic acids and $SO_2$ is not really new. Other studies have found the formation of an acid and the release of $SO_2$ from the secondary ozonide as the most energetically favourable reaction channel for even smaller sCI, (Kurten, 2011; Vereecken, 2012). The experimental results obtained in this study support those previous theoretical works.

"Table 1 does not indicate which experiments include water":

All the experiment summarised in Table 1 were carried out with HR=0. It was stated in page 7, line 10.  For sake of clarity now this information is included also in the table in the revised manuscript.

As suggested by the referee, the energy of the reactants (alkene + $SO_2$) and the primary ozonide have been included in table 2 and in figure 5.  Due to the exothermic nature of the ozonolysis, unimolecular decomposition and stabilisation through collision would compete in the formation of sCI. A comment in this sense has been included in the manuscript.

Concerning to water concentrations, we have  RH meters but their ranges are not useful for RH below 1%.  The stated water ratio from the cylinder is 1-2ppm (Praxair). We have tried to calculate the water concentration in the synthetic air after drying it through a $LN_2$ trap with molecular sieve 5A using a mass spectrometer but the residual $H_2O$ signal from chamber degas is too high to enable quantification. We have also tried with a FTIR but the purge instability in the FTIR was above the expected changes in the water signal.

From the literature-available data for ice, passing the synthetic air through the $LN_2$ trap, the vapour pressure is expected to fall well below $10^{-7}$Pa (Murphy and Koop, 2005) and thus the expected concentration in the dried synthetic air would be below $1x10^7$molecule $cm^{-3}$.

Since residual concentrations could be higher, we have conducted a series of experiments to test the possibility of $SO_3$–water reaction in the reactor and to estimate the water concentration. We have used a degassed sample of solid sulfur trioxide (99.5%, stabilized, Aldrich) contained in a glass flask to obtain different $SO_3$ concentrations in the reactor (as it was done in previous studies (Jayne, 1997).  Freshly dried synthetic air was mixed with $SO_3$ in the teflon reactor and the mixture was continuously monitored by the CPC for 40 minutes. The figure shows the results for experiments with $SO_3$ concentration in the range 1 to 12ppb. No particles could be observed for experiments with low initial $SO_3$ concentrations. On the other hand, NPF was observed for the experiments with $SO_3$ in the range 6 to 12 ppb. Under these experimental conditions, nucleation is attributed to the formation of $H_2SO_4$ from the reaction of $SO_3$ and the residual $H_2O$.

The overall  gas-phase reaction $H_2O + SO_3 \rightarrow H_2SO_4$  exhibits a second-order dependence on water vapor concentration, the first-order rate coefficient for the $SO_3$ loss being k= $3.90x10^{-41}$exp(6830.6/T)$[H_2O]^2$ (Jayne, 1997).

Taking into account that the approximate $H_2SO_4$ gas phase concentration able to nucleate is around $5x10^6$molecule $cm^{-3}$ (Metzger, 2010), the concentration of water in the reactor may be obtained by simulating the $SO_3$ and $H_2SO_4$ profiles for different guessed $H_2O$ profiles.

Thus for example, since no NPF was observed for the experiment with 2ppb of $SO_3$, the water concentration must be below 15ppb. On the other hand from the experiment with 6ppb of $SO_3$, a 20 ppb water concentration is required to reproduce the observed nucleation.

From all the experiments carried out, we estimate that water concentration in the reactor is 20±10ppb. To check for permeation through the reactor wall, some experiments have been also carried out with dry air after 1 hour in the reactor. The results were similar to those carried out with freshly dried air.

[Figure]

This figure and the related discussion has been included in the supporting information.

According to the $SO_2$ comments, direct measurements of $SO_2$ are included in this reply and in the revised manuscript. Furthermore, these weeks we had an ozone analyser available that provided additional information concerning the gas phase (Figure 2). The experimental and simulated profiles of ozone (based on the literature ozonolysis rate constant) are in good agreement and so ozone is not quantitatively lost through other reactions.

[Figure]

We have carried out new experiments with lower $SO_2$ concentrations in the range 10-20ppb. In all cases the profile of $SO_2$ remained neatly constant. For example it was 10.0±0.2 ppb during the whole experiment for the experiments with 10ppb of $SO_2$. A first order loss rate constant may be derived from the $SO_2$ profile: $k = 8.5 \times 10^{-6} s^{-1}$. Thus, $SO_2$ losses (if they occur) are very low. See the figure.

[Figure]

To check the possibility of $SO_3$ production, we can assume a simple mechanism where any lost $SO_2$ molecule would be converted exclusively into $SO_3$:

$SO_2 \rightarrow SO_3$ $\quad\quad\quad$ k= $8.5 \times 10^{-6} s^{-1}$ $\quad\quad$ $[SO_2]_o$ = 10ppb

And then $SO_3$ would exclusively react with water to produce $H_2SO_4$:

$H_2O + SO_3 \rightarrow H_2SO_4$ $\quad\quad$ k= $3.90 \times 10^{-41} exp(6830.6/T)[H2O]^2$ $\quad$ (Jayne, 1997)

Simulating the $SO_2$, $SO_3$ and $H_2O$ profiles for a 20ppb water concentration and for 10ppb initial $SO_2$ concentration, it would require more than 1 hour to generate $5 \times 10^6$ molecule $cm^{-3}$ of $H_2SO_4$, which is the approximate concentration able to nucleate (Metzger, 2010). For 20ppb initial $SO_2$ concentration it would require 28 minutes. Nevertheless, for these experiment nucleation was visible at 2minutes (almost instantaneous if we take away the mixing time of reactants).

Thus, considering the low levels of water vapour in the reactor and the observation of nearly constant $SO_2$ for the experiments with lower $SO_2$ concentrations, the contribution of $SO_3$-$H_2SO_4$ pathway to NPF seems to be minor and unable to lead to nucleation by itself. In this sense the catalytic pathway releasing $SO_2$, which is thermodynamic- favourable, may be the key to NPF.

**Main changes in the manuscript.**

Page 1, line 14. It has been rewritten. Water presence at ppb-ppm concentration may have an effect on SOA production. Nevertheless, for higher concentrations, no effect was found.

Page 1, line 18. $SO_3$ role is not overall ruled out. $SO_2$ catalysed reactions are suggested as an additional pathway to NPF.

Page 1, line 18. $SO_3$ is not excluded as a possible intermediate to produce SOA.

Page 3, line 31. The term "completely dry conditions" has been removed.

Page 4, line 4. The estimated water concentration inside the reactor is stated.

Page 4, line 22. The experiments carried out with the ozone analyser are introduced.

Page 6. Line 9. New data and discussion about $SO_2$, $SO_3$ and sulfuric acid is introduced.

Page 7, line 21. New data and discussion about $SO_2$, $SO_3$ and sulfuric acid is introduced.

Page 9, line 29. The energy of the first step of the ozonolysis is introduced.

Page 10, line 18. From the results of this study, (R4) is suggested as the probable pathway to NPF.

Page 10, line 24. Vapour pressure estimates are given.

Page 11, line 14. $SO_3$ role is not overall ruled out. The statements concerning $SO_2$ are restricted to low $SO_2$ concentration conditions.

Table 2. The optimized energies of the reactants and first ozonides are included in the table.

Figure 2 includes experimental gas-phase profiles for $SO_2$ and $O_3$.

Figure 3 includes the $O_3$ experimental profile.

Figure 5a. Reactants and the first ozonides have been included in the mechanism scheme.

New figures, S1 and S3, have been introduced in the supporting information.

**References:**

Jayne, J. T., Po1schl, U., Chen, Y., Dai, D., Molina, L.T., Worsnop, D.R., Kolb, C.E., Molina, M. J.: Pressure and Temperature Dependence of the Gas-Phase Reaction of SO3 with H2O and the Heterogeneous Reaction of SO3 with H2O/H2SO4 Surfaces. J. Phys. Chem. A, 101, 10000-10011, 1997.

Kurtén, T., Lane, J. R., Jørgensen, S., Kjaergaard, H. G.: A Computational Study of the Oxidation of $SO_2$ to $SO_3$ by Gas-Phase Organic Oxidants. J. Phys. Chem. A,116, 6823-6830, 2011.

Metzger, A., Verheggen, B., Dommen, J., Duplissy, J., Prevota, A. S. H., Weingartner, E.,Riipinen, I, Kulmala, M., Spracklend, D. V., Carslaw, K. S., Baltensperger, U.: Evidence for the role of organics in aerosol particle formation under atmospheric conditions. PNAS. 107, 6646-6651, 2010.

Murphy, D., M. and Koop, T.: Review of the vapour pressures of ice and supercooled water for atmospheric applications, Quart. J. Royal Meterol. Soc., 131, 1539–1565, 2005.

Vereecken, L., Harder, H., Novelli, A.: The reaction of Criegee intermediates with NO, $RO_2$, and $SO_2$, and their fate in the atmosphere, Phys. Chem. Chem. Phys., 14, 14682–14695, 2012.

**Anonymous Referee #2**

This paper represents a new chemical mechanism potentially relevant for NPF, but the experimental data does not fully prove the proposed mechanism. Lack of experimental support is compensated by theoretical arguments and I guess this paper could be published once the problems pointed out by referee #1 and my concerns below are addressed.

Reviewer #1 already provided an extensive and complete review and I avoid repeating that but I still want to express my major concerns despite the overlap. My main concern is that authors quite vaguely exclude the SO3 channel and the role of sulphuric acid in observed NPF. Proper exclusion of SO3 channel is critical since authors are providing a new chemical mechanism arising from SO3 exclusion and explaining NPF in their system with previously unrecognised pathways. I cannot judge myself, if the proposed mechanism with SO2 catalyzing Criegee conversion to organic acids or aldehydes is

relevant or not, but I do question a) the absence of sulphuric acid in the system and b) the atmospheric relevance of the proposed mechanism even if correct and relevant in chamber conditions.

Authors state that reaction of SO3 with water vapour cannot be related to NPF since there is no water. Water is omnipresent even in authors' chamber and the reaction of SO3 with water is fast and unlikely seriously limited by the availability of water even in "dry" conditions. Water can come from the walls, and even through the Teflon wall, with trace gases and from the synthetic air bottle. Even if the lack of water would slow down the SO3 conversion to sulphuric acid, vast amounts of SO3 can be produced from sCI+SO2 or OH+SO2 and minute water concentrations could be enough for sufficient sulphuric acid formation. What are the yields of OH from ozonolysis of these DHFs? OH scavenging was >95%. Still, with very high concentrations of furan and ozone, the OH production could potentially be high enough that the residual <5% reacting with SO2 (High concentrations up to 0.5ppm!) can be a significant source of SO3 and subsequently sulphuric acid under assumption that water residuals are present. More is required to show that not enough sulphuric acid can be formed via sCI+SO2/OH+SO2 -> SO3 (+H2O) -> H2SO4. (Even though the presented theoretical analyzis suggests SO3 is not released from the reaction of sCI and SO2, previous literature shows that's unlikely the case with most alkenes). With some assessment of water vapour concentration upper limit and with known or approximated reaction rates and yields authors could maybe get at least an idea about maximum sulphuric acid concentrations in the chamber.

The statement that SO2 remained constant (p6., l7.) is not supported by data shown. Was it measured? And if it was measured, with 0.5 ppm SO2, there's 1e13 molecules of it in a cm-3. If one per-mille of that is converted to H2SO4, that would be sufficient (concentration up to some 1e10 molec cm-3) to drive NPF with unavoidable background contaminants (e.g. ammonia or amines) or with some products from DHF oxidation and produce the observed NPF rate of maybe 1000 #/s. And a 1 per-mil drop

in SO2 may be tricky to observe. So what is really the experimental evidence for the SO2 recycling?

All in all, before publishing this paper, I would like to see more results and discussion to exclude the SO3 channel and sulphuric acid produced via that channel either purely from OH or with an assumption that theoretical prediction of no SO3 formation from sCI was incorrect. If, from experimental data, the presence of sulphuric acid cannot be excluded, the paper should be written in a manner that accounts for that deficiency.

I also have some doubts that any organic acids formed from these relatively small alkenes would have sufficiently low vapour pressures that they could homogeneously nucleate. (There are also methods to estimate the vapour pressure of such compounds, see e.g. Donahue et al. Atmos Chem Phys 11, 3303–3318, 2011; Pankow et al., Atmos Chem Phys 8, 2773–2796, 2008). These acids may be partitioning between particle/wall and gas phase, but that they underwent homogeneous nucleation with high nucleation rates is more questionable. If authors suggest nucleation is driven by proposed compounds, some more data or discussion would be appreciated.

On the other hand, the concentrations in the chamber system are vastly above the atmospheric ones (ozone, DHF and SO2 are 100-100 times higher than typical for the atmosphere), meaning that at least the atmospheric relevance of such acids, even if they were nucleating in the chamber, is more than questionable and against the present understanding on atmospheric nucleation processes.

Intro: Authors largely exclude the discussion related to extremely low volatile organics formed in auto-oxidation reactions from alkene ozonolysis (Ehn et al., Nature, 506, 476-, Nature; and many subsequent publications) considered to be one of the main pathways to atmospheric NPF and SOA. Since the title and motivation of this paper is

SOA, the major progress on that field should be shortly discussed in the introduction.

**Answer to referee #2.**

We sincerely thank the comments of the referee. Additional information is now included together with data from new experiments carried out during these few weeks in line with the requirements of the referee.

The mechanism for sCI reaction with $SO_2$ to generate organic acids and $SO_2$ is not really new. Other studies have found the formation of an acid and the release of $SO_2$ from the secondary ozonide as the most energetically favourable reaction channel for even smaller sCI (Jiang, 2010; Kurten, 2011; Vereecken, 2012). The experimental results obtained in this study are consistent with the isomerisation channel proposed in these previous theoretical works.

**a)"Absence of sulphuric acid in the system".**

From the literature-available data for ice, passing the synthetic air through the $LN_2$ trap, the vapour pressure is expected to fall well below $10^{-7}$Pa (Murphy and Koop, 2005) and thus the expected concentration in the dried synthetic air is below $1x10^7$molecule $cm^{-3}$.

Since residual concentrations could be higher, we have conducted a series of experiments to test the possibility of $SO_3$–water reaction in the reactor and to estimate the water concentration. We have used a degassed sample of solid sulfur trioxide (99.5%, stabilized, Aldrich) contained in a glass flask to obtain different $SO_3$ concentrations in the reactor (as it was done in previous studies (Jayne, 1997).  Freshly dried synthetic air was mixed with $SO_3$ in the teflon reactor and the mixture was continuously monitored by the CPC for 40 minutes. The figure shows the results for experiments with $SO_3$ concentration in the range 1 to 12ppb. No particles could be observed for experiments with low initial $SO_3$ concentrations. On the other hand, NPF was observed for the experiments with $SO_3$ in the range 6 to 12 ppb. Under these experimental conditions, nucleation is attributed to the formation of $H_2SO_4$ from the reaction of $SO_3$ and the residual $H_2O$.

 The overall  gas-phase reaction $H_2O + SO_3 \rightarrow H_2SO_4$   exhibits a second-order dependence on water vapor concentration, the first-order rate coefficient for the $SO_3$ loss being k= $3.90x10^{-41}$exp(6830.6/T)$[H2O]^2$ (Jayne, 1997).

Taking into account that the approximate $H_2SO_4$ gas phase concentration able to nucleate is around $5x10^6$molecule $cm^{-3}$ (Metzger, 2010), the concentration of water in the reactor may be obtained by simulating the $SO_3$ and $H_2SO_4$ profiles for different guessed $H_2O$ profiles.

Thus for example, since no NPF was observed for the experiment with 2ppb of $SO_3$, the water concentration must be below 15ppb. On the other hand from the experiment with 6ppb of $SO_3$, a 20ppb water concentration is required to reproduce the observed nucleation.

From all the experiments carried out, we estimate that water concentration in the reactor is 20±10ppb. To check for permeation through the reactor wall, some experiments have been also carried out with dry air after 1 hour in the reactor. The results were similar to those carried out with freshly dried air.

[Figure]

This figure and the related discussion has been included in the supporting information.

According to the $SO_2$ comments, direct measurements of $SO_2$ are included in this reply and in the revised manuscript. We have carried out new experiments with lower $SO_2$ concentrations in the range 10-20ppb. In all cases the profile of $SO_2$ remained neatly constant. For example it was $10.0\pm0.2$ppb during the whole experiment for the experiments with 10ppb of $SO_2$. A first order loss rate constant may be derived from the $SO_2$ profile: $k= 8.5\times10^{-6}s^{-1}$. Thus, $SO_2$ losses (if they occur) are very low. See the figure.

[Figure]

To check the possibility of $SO_3$ production, we can assume a simple mechanism where any lost $SO_2$ molecule would be converted exclusively into $SO_3$:

$SO_2 \rightarrow SO_3$          $k= 8.5 \times 10^{-6} s^{-1}$        $[SO_2]_o = 10 ppb$

And then $SO_3$ would exclusively react with water to produce $H_2SO_4$:

$H_2O + SO_3 \rightarrow H_2SO_4$      $k= 3.90 \times 10^{-41} exp(6830.6/T)[H2O]^2$      (Jayne, 1997)

Simulating the $SO_2$, $SO_3$ and $H_2O$ profiles for a 20ppb water concentration and for 10ppb initial $SO_2$ concentration, it would require more than 1 hour to generate $5 \times 10^6$ molecule $cm^{-3}$ of $H_2SO_4$, which is the approximate concentration able to nucleate (Metzger, 2010). For 20ppb initial $SO_2$ concentration it would require 28 minutes. Nevertheless, for these experiment nucleation was visible at 2minutes (almost instantaneous if we take away the mixing time of reactants).

Relatively high levels of OH would deplete $SO_2$ if $SO_2$ concentration were very low. In this sense, the experiments with high dihydrofurans and ozone concentrations (for example 0.5 and 1.0 ppm, respectively) and low $SO_2$ concentration (in the range of 10ppb) show that residual OH concentration must be negligible in the system since the experimental $SO_2$ concentration remained constant during the experiments.

Thus, considering the low levels of water vapour in the reactor and the observation of nearly constant $SO_2$ for the experiments with lower $SO_2$ concentrations, the contribution of $SO_3$-$H_2SO_4$ pathway to NPF seems to be minor and unable to lead to nucleation by itself. In this sense the catalytic pathway releasing $SO_2$, which is thermodynamic-favourable, may be the key to NPF.

For higher $SO_2$ concentrations (in the range of 0.5 ppm) small changes at the level of the uncertainty of the $SO_2$ measurements can not be completely excluded as a possible source of $SO_3$.

**b)Atmospheric relevance.**

We have carried out experiments at lower concentrations. Thus, for example, for 2,5-DHF no particles were detected for 0.02, 0.04 and 0.02ppm concentrations of 2,5-DHF, ozone and $SO_2$, respectively. For 0.05, 0.1 and 0.05 ppm concentrations (2 ,5-DHF, ozone and $SO_2$) NPF could be observed but particle number concentration, particle size diameter and particle mass concentration were very low and noisy and could not be measured accurately. To assess the effects of water, $SO_2$ and ozone, the concentrations had to be increased.

Although the concentrations of reactants in this study are higher than average concentrations in the atmosphere and nucleation from the ozonolysis of only DHFs is not expected, this work shows that these reaction lead to condensable species that could contribute to NPF or particle growing in the atmosphere. Furthermore this study reports theoretical and experimental data that points the catalytic role of $SO_2$ in the oxidation of SCIs.

**Vapour pressure.**

From the work of Donahue et al (2011) a saturation mass concentration around 100 microgram.$m^{-3}$ is expected for C4 chemicals with a 1:1 oxygenation ratio (O:C). It is the high degree of oxygenation which leads to a low volatility value, $5 \times 10^{11}$ molecules $cm^{-3}$. For the range of initial reactant concentrations in the laboratory experiments this range of product concentration could be reached. A comment has been introduced in the manuscript as suggested by the referee.

**Ehn 2014 reference** has been included in the introduction as suggested by the referee.

**Main changes in the manuscript.**

Page 1, line 14. It has been rewritten. Water presence at ppb-ppm concentration may have an effect on SOA production. Nevertheless, for higher concentrations, no effect was found.

Page 1, line 18. $SO_3$ role is not overall ruled out. $SO_2$ catalysed reactions are suggested as an additional pathway to NPF.

Page 1, line 18. $SO_3$ is not excluded as a possible intermediate to produce SOA.

Page 3, line 31. The term "completely dry conditions" has been removed.

Page 4, line 4. The estimated water concentration inside the reactor is stated.

Page 4, line 22. The experiments carried out with the ozone analyser are introduced.

Page 6. Line 9. New data and discussion about $SO_2$, $SO_3$ and sulfuric acid is introduced.

Page 7, line 21. New data and discussion about $SO_2$, $SO_3$ and sulfuric acid is introduced.

Page 9, line 29. The energy of the first step of the ozonolysis is introduced.

Page 10, line 18. From the results of this study, (R4) is suggested as the probable pathway to NPF.

Page 10, line 24. Vapour pressure estimates are given.

Page 11, line 14. $SO_3$ role is not overall ruled out. The statements concerning $SO_2$ are restricted to low $SO_2$ concentration conditions.

Table 2. The optimized energies of the reactants and first ozonides are included in the table.

Figure 2 includes experimental gas-phase profiles for $SO_2$ and $O_3$.

Figure 3 includes the $O_3$ experimental profile.

Figure 5a. Reactants and the first ozonides have been included in the mechanism scheme.

New figures, S1 and S3, have been introduced in the supporting information.

**References:**

Jayne, J. T., Po1schl, U., Chen, Y., Dai, D., Molina, L.T., Worsnop, D.R., Kolb, C.E., Molina, M. J.: Pressure and Temperature Dependence of the Gas-Phase Reaction of SO3 with H2O and the Heterogeneous Reaction of SO3 with H2O/H2SO4 Surfaces. J. Phys. Chem. A, 101, 10000-10011, 1997.

Kurtén, T., Lane, J. R., Jørgensen, S., Kjaergaard, H. G.: A Computational Study of the Oxidation of $SO_2$ to $SO_3$ by Gas-Phase Organic Oxidants. J. Phys. Chem. A,116, 6823-6830, 2011.

Metzger, A., Verheggen, B., Dommen, J., Duplissy, J., Prevota, A. S. H., Weingartner, E.,Riipinen, I, Kulmala, M., Spracklend, D. V., Carslaw, K. S., Baltensperger, U.: Evidence for the role of organics in aerosol particle formation under atmospheric conditions. PNAS. 107, 6646-6651, 2010.

Murphy, D., M. and Koop, T.: Review of the vapour pressures of ice and supercooled water for atmospheric applications, Quart. J. Royal Meterol. Soc., 131, 1539–1565, 2005.

Vereecken, L., Harder, H., Novelli, A.: The reaction of Criegee intermediates with NO, $RO_2$, and $SO_2$, and their fate in the atmosphere, Phys. Chem. Chem. Phys., 14, 14682–14695, 2012.

**Anonymous Referee #3**

The authors describe experimental findings from the ozonolysis reaction of 2,3- and 2,5-dihydrofuran at atmospheric pressure and room temperature. Experiments were carried out in a Teflon bag with special attention to SOA formation. Particle formation was followed by total number measurements using a TSI CPC 3775 as well by measuring the particle size distribution by means of a TSI FMPS 3091. Gas-phase species, such as ozone, the dihydrofurans and water vapor, were not monitored in the course of the reaction. For runs in presence of SO2, the SO2 time series were only monitored "For some experiments" but no information on that is presented in the manuscript. The authors concluded as a result of their experiments that i) the detected particle formation was not connected to sulfuric acid, ii) the reaction of Criegee intermediates (CI) with SO2 proceeds without SO3 formation and SO2 serves as a "catalyst" for acid formation starting from CI. These are very interesting statements. But unfortunately,

the experiments do not distinctly support the conclusions. A significant fraction of other products than sulfuric acid from CI+SO2 would be very important for the understanding of CI0s role in atmospheric oxidation.
Here my critical points:
- The authors used very high reactant concentrations, far away from atmospheric conditions. Why they are doing so? High initial concentrations connected with high intermediate concentrations can open reaction channels not relevant for the atmosphere. Particle measurements are sensitive enough allowing to work close to atmospheric reactant conditions.
- Cyclohexane was used as OH scavenger for more than 95% of the OH radicals. But what about the residual OH radicals? They are definitely reacting with SO2 in competition with all other OH reactions in the system forming finally sulfuric acid.
- What does it mean "completely dry conditions"? A measurement of the water vapor concentration in the Teflon bag is needed. It0s very challenging to produce and handle an extremely dry reaction gas with a water vapor concentration as low as needed that hydrolysis of SO3 doesn0t work!
- Sulfuric acid measurements are needed in order to rule out a significant contribution of sulfuric acid for nucleation and particle growth. I would encourage the authors to collaborate with groups familiar with the H2SO4 CIMS technique.
- The authors should show the SO2 time series or at least an example. What0s the uncertainty of the SO2 detection? I guess, from a constant SO2 time series of 10(12) molecules cm(-3) or more it is impossible concluding that 10(9) or 10(10) molecules cm(-3) have been converted. And that0s enough to produce the needed sulfuric acid for nucleation and early growth.
All together, I guess this manuscript needs a major revision based on additional experiments. It could become an important paper in the field of CI reactions. From my perspective, at the moment the experimental basis is not good enough.

**Answer to referee #3.**

We sincerely thank the comments of the referee. Additional information is now included together with data from new experiments carried out during these few weeks in line with the requirements of the referee.

**High reactant concentrations.**

We have carried out experiments at lower concentrations. Thus, for example, for 2,5-DHF no particles were detected for 0.02, 0.04 and 0.02 ppm concentrations of 2,5-DHF, ozone and $SO_2$, respectively. For 0.05, 0.1 and 0.05 ppm concentrations (2,5-DHF, ozone and $SO_2$) NPF could be observed but particle number concentration, particle size diameter and particle mass concentration were very low and noisy and could not be measured accurately. To assess the effects of water, $SO_2$ and ozone, the concentrations had to be increased.

**"Completely dry conditions" and sulphuric acid measurements**.

This term has been removed from the manuscript. For the experiments with dried synthetic air (passing through a liquid nitrogen trap), the concentration of water in the reactor has been estimated. We have conducted a series of experiments to test the possibility of $SO_3$ –water

reaction in the reactor and to estimate the water concentration. We have used a degassed sample of solid sulfur trioxide (99.5%, stabilized, Aldrich) contained in a glass flask to obtain different $SO_3$ concentrations in the reactor (as it was done in previous studies (Jayne, 1997). Freshly dried synthetic air was mixed with $SO_3$ in the teflon reactor and the mixture was continuously monitored by the CPC for 40 minutes. The figure shows the results for experiments with $SO_3$ concentration in the range 1 to 12ppb. No particles could be observed for experiments with low initial $SO_3$ concentrations. On the other hand, NPF was observed for the experiments with $SO_3$ in the range 6 to 12 ppb. Under these experimental conditions, nucleation is attributed to the formation of $H_2SO_4$ from the reaction of $SO_3$ and the residual $H_2O$.

The overall gas-phase reaction $H_2O + SO_3 \rightarrow H_2SO_4$ exhibits a second-order dependence on water vapor concentration, the first-order rate coefficient for the $SO_3$ loss being k= $3.90 \times 10^{-41} exp(6830.6/T)[H2O]^2$ (Jayne, 1997).

Taking into account that the approximate $H_2SO_4$ gas phase concentration able to nucleate is around $5 \times 10^6$ molecule cm$^{-3}$ (Metzger, 2010), the concentration of water in the reactor may be obtained by simulating the $SO_3$ and $H_2SO_4$ profiles for different guessed $H_2O$ profiles.

Thus for example, since no NPF was observed for the experiment with 2ppb of $SO_3$, the water concentration must be below 15ppb. On the other hand from the experiment with 6ppb of $SO_3$, a 20ppb water concentration is required to reproduce the observed nucleation.

From all the experiments carried out, we estimate that water concentration in the reactor is $20\pm10$ppb. To check for permeation through the reactor wall, some experiments have been also carried out with dry air after 1 hour in the reactor. The results were similar to those carried out with freshly dried air.

[Figure]

This figure and the related discussion has been included in the supporting information.

**$SO_2$ time series.**

According to the $SO_2$ comments, direct measurements of $SO_2$ are included in this reply and in the revised manuscript.

We have also carried out new experiments with lower $SO_2$ concentrations in the range 10-20ppb. In all cases the profile of $SO_2$ remained neatly constant. For example it was 10.0±0.2 ppb during the whole experiment for the experiments with 10ppb of $SO_2$. A first order loss rate constant may be derived from the $SO_2$ profile: $k= 8.5 \times 10^{-6} s^{-1}$. Thus, $SO_2$ losses (if they occur) are very low. See the figure.

[Figure]

To check the possibility of $SO_3$ production, we can assume a simple mechanism where any lost $SO_2$ molecule would be converted exclusively into $SO_3$:

$SO_2 \rightarrow SO_3$ $\qquad k= 8.5 \times 10^{-6} s^{-1}$ $\qquad [SO_2]_o = 10$ppb

And then $SO_3$ would exclusively react with water to produce $H_2SO_4$:

$H_2O + SO_3 \rightarrow H_2SO_4$ $\qquad k= 3.90 \times 10^{-41} \exp(6830.6/T)[H2O]^2$ (Jayne, 1997)

Simulating the $SO_2$, $SO_3$ and $H_2O$ profiles for a 20ppb water concentration and for 10ppb initial $SO_2$ concentration, it would require more than 1 hour to generate $5 \times 10^6$ molecule $cm^{-3}$ of $H_2SO_4$, which is the approximate concentration able to nucleate (Metzger, 2010). For 20ppb initial $SO_2$ concentration it would require 28 minutes. Nevertheless, for these experiment nucleation was visible at 2minutes (almost instantaneous if we take away the mixing time of reactants).

Thus, considering the low levels of water vapour in the reactor and the observation of nearly constant $SO_2$ for the experiments with lower $SO_2$ concentrations, the contribution of $SO_3$-$H_2SO_4$ pathway to NPF seems to be minor and unable to lead to nucleation by itself. In this sense the catalytic pathway releasing $SO_2$, which is thermodynamic- favourable, may be the key to NPF.

**Cyclohexane - OH excavenger.**

Relatively high levels of OH would deplete $SO_2$ if $SO_2$ concentration were very low. Nevertheless, even in those experiments where OH level could be higher (the experiments with dihydrofurans and ozone concentrations in the range of 0.5 and 1.0 ppm, respectively) and low $SO_2$ concentration (in the range of 10ppb), the $SO_2$ concentration did not fall during

the experiment. These results suggest that that residual OH concentration must be negligible in the system.

**Main changes in the manuscript.**

Page 1, line 14. It has been rewritten. Water presence at ppb-ppm concentration may have an effect on SOA production. Nevertheless, for higher concentrations, no effect was found.

Page 1, line 18. $SO_3$ role is not overall ruled out. $SO_2$ catalysed reactions are suggested as an additional pathway to NPF.

Page 1, line 18. $SO_3$ is not excluded as a possible intermediate to produce SOA.

Page 3, line 31. The term "completely dry conditions" has been removed.

Page 4, line 4. The estimated water concentration inside the reactor is stated.

Page 4, line 22. The experiments carried out with the ozone analyser are introduced.

Page 6. Line 9. New data and discussion about $SO_2$, $SO_3$ and sulfuric acid is introduced.

Page 7, line 21. New data and discussion about $SO_2$, $SO_3$ and sulfuric acid is introduced.

Page 9, line 29. The energy of the first step of the ozonolysis is introduced.

Page 10, line 18. From the results of this study, (R4) is suggested as the probable pathway to NPF.

Page 10, line 24. Vapour pressure estimates are given.

Page 11, line 14. $SO_3$ role is not overall ruled out. The statements concerning $SO_2$ are restricted to low $SO_2$ concentration conditions.

Table 2. The optimized energies of the reactants and first ozonides are included in the table.

Figure 2 includes experimental gas-phase profiles for $SO_2$ and $O_3$.

Figure 3 includes the $O_3$ experimental profile.

Figure 5a. Reactants and the first ozonides have been included in the mechanism scheme.

New figures, S1 and S3, have been introduced in the supporting information.

**References:**

Jayne, J. T., Po1schl, U., Chen, Y., Dai, D., Molina, L.T., Worsnop, D.R., Kolb, C.E., Molina, M. J.: Pressure and Temperature Dependence of the Gas-Phase Reaction of SO3 with H2O and the Heterogeneous Reaction of SO3 with H2O/H2SO4 Surfaces. J. Phys. Chem. A, 101, 10000-10011, 1997.

Kurtén, T., Lane, J. R., Jørgensen, S., Kjaergaard, H. G.: A Computational Study of the Oxidation of SO2 to SO3 by Gas-Phase Organic Oxidants. J. Phys. Chem. A,116, 6823-6830, 2011.

Metzger, A., Verheggen, B., Dommen, J., Duplissy, J., Prevota, A. S. H., Weingartner, E.,Riipinen, I, Kulmala, M., Spracklend, D. V., Carslaw, K. S., Baltensperger, U.: Evidence for the role of organics in aerosol particle formation under atmospheric conditions. PNAS. 107, 6646-6651, 2010.

Murphy, D., M. and Koop, T.: Review of the vapour pressures of ice and supercooled water for atmospheric applications, Quart. J. Royal Meterol. Soc., 131, 1539–1565, 2005.

Vereecken, L., Harder, H., Novelli, A.: The reaction of Criegee intermediates with NO, $RO_2$, and $SO_2$, and their fate in the atmosphere, Phys. Chem. Chem. Phys., 14, 14682–14695, 2012.